# Coevolution of male and female mate choice can destabilize reproductive isolation

Thomas G. Aubier [1,2]*, Hanna Kokko [2] & Mathieu Joron [1]*

Sexual interactions play an important role in the evolution of reproductive isolation, with important consequences for speciation. Theoretical studies have focused on the evolution of mate preferences in each sex separately. However, mounting empirical evidence suggests that premating isolation often involves mutual mate choice. Here, using a population genetic model, we investigate how female and male mate choice coevolve under a phenotype matching rule and how this affects reproductive isolation. We show that the evolution of female preferences increases the mating success of males with reciprocal preferences, favouring mutual mate choice. However, the evolution of male preferences weakens indirect selection on female preferences and, with weak genetic drift, the coevolution of female and male mate choice leads to periodic episodes of random mating with increased hybridization (deterministic 'preference cycling' triggered by stochasticity). Thus, counterintuitively, the process of establishing premating isolation proves rather fragile if both male and female mate choice contribute to assortative mating.

[1] Centre d'Ecologie Fonctionnelle et Evolutive, CEFE - UMR 5175 - CNRS, Université de Montpellier, EPHE, Université Paul Valéry, 1919 route de Mende, F-34293 Montpellier 5, France. [2] Department of Evolutionary Biology and Environmental Studies, University of Zurich, Winterthurerstrasse 190, CH-8057 Zurich, Switzerland. *email: thomas.aubier@normalesup.org; mathieu.joron@cefe.cnrs.fr

Reproductive isolation among taxa can be caused by different isolating barriers, such as ecological divergence, sexual isolation, hybrid sterility, and microspatial partitioning[1,2]. While those barriers have been well described, their temporal stability is little studied either theoretically or empirically. Yet, if the barriers that isolate diverged populations are not stable, the consequent recurring hybridization may have profound consequences for species evolution (e.g., speciation[3], transgressive segregation[4], adaptive introgression[5], and genetic swamping[6]).

Assortative mating—the tendency of individuals of similar phenotype to mate together more often than expected by chance—is widespread in animals[7] and plays a key role in generating premating reproductive isolation[8]. Assortative mating can arise as a by-product of adaptive divergence via temporal or spatial isolation[9], or can be driven by various behavioral processes[10]. Of particular interest is the case of homotypic mate preferences ("matching mating rule"[11]), where individuals preferentially choose mates with which they share phenotypic traits such as colors[12,13] or acoustic signals[14]. When these traits are simultaneously under divergent or disruptive ecological selection (so-called "magic traits"[15,16]), choosy individuals with homotypic mate preferences are less likely to produce offspring with unfit intermediate phenotypes. Theoretically, this effect creates indirect selection favoring a further increase in choosiness (i.e., stronger homotypic preferences), establishing, or strengthening premating isolation between diverging populations[11,17–22]. In addition, choosiness often induces positive frequency-dependent sexual selection that favors the most common phenotype; if mating success varies among individuals and assortment is not perfect, individuals having the most common phenotype have the highest mating success. Once diverging populations are sufficiently differentiated with respect to a magic trait, disruptive ecological selection on that trait is therefore complemented by disruptive sexual selection due to choosiness, which in turn can drive the evolution of even stronger choosiness[23–25] (but note that this sexual selection pressure can also inhibit the initial evolution of strong choosiness in sympatry[24–26] or parapatry[16,27,28]). Empirically, homotypic preferences based on "magic traits" subject to disruptive selection are, indeed, often associated with premating reproductive isolation during speciation in the presence of gene flow[16,29,30]. Here, we are interested in whether this isolating barrier is stable when mate choice can be expressed by both sexes.

Previous theoretical developments have focused on the evolution of choosiness in each sex separately[17–22,31,32]. In principle, the indirect selection effect explained above (i.e., reduced production of unfit intermediate offspring) favors increased choosiness in both females and males. Nevertheless, evolutionary pressures acting on the two sexes are profoundly different. Under idealized conditions of polygyny and unlimited male mating potential, males can be thought of as an unlimited resource for females and all females have therefore equal mating success[33]. Females, on the other hand, often represent a limited resource for males, with male–male competition for access to females generating differential male mating success. Consequently, having a preference directly affects how much competition a male faces for gaining a mate. Typically, males place themselves in a disadvantageous competitive setting if they preferentially court "popular" females—that can be phrased as sexual selection acting directly against male preferences[31,34,35].

The extent of courtship effort (resources[36,37] or time[38]) allocated toward preferred females is another key, but an underappreciated, factor in the evolution of male mate choice. Male preferences can only evolve if the lack of mating attempts with unpreferred females improves mating opportunities with preferred ones. Typically, this is conceptualized as reallocation of courtship effort, and male mate choice is hampered if complete reallocation is difficult to achieve. Male preferences can nevertheless evolve if direct or indirect benefits (e.g., increased probability to mate[34,39], fertility[34,40], or offspring quality[31,41]) outweigh these costs. Previous theoretical studies have shown that male choosiness can evolve by reinforcement[31], and that strategic male courtship allocation can generate polymorphic male preferences[32], which may ultimately lead to reproductive isolation between populations. Overall, however, female choosiness is considered more likely to evolve than male choosiness, at least as long as it does not associate with competitive and opportunity costs[31].

In some interspecific sexual interactions, both males and females discriminate against heterospecifics, and therefore engage in mutual mate choice with respect to species identity[42–45]. In cichlid fishes[46,47] and Heliconius butterflies[2,13,48–50], which are textbook examples of potential speciation via premating isolation, both males and females can display homotypic preferences based on color. However, the consequences of mutual mate preferences for reproductive isolation remain to be explored. Preferences have been shown to evolve independently if female and male choices are based on distinct traits[34]. However, females and males with mutual homotypic preferences often use the same trait to evaluate potential mates. Choosiness in one sex therefore influences the evolution of choosiness in the other through genetic linkage disequilibrium[34,51–54]. Female choosiness may also strongly favor the evolution of male choosiness by directly increasing the mating success of choosy males focusing their courtship effort on females that are likely to accept them as mates.

Here, by analyzing a population genetic model, we characterize the coevolutionary dynamics of female and male mate choice based on the same phenotypic trait under disruptive selection. We then assess its effects on the stability of reproductive isolation. We show that female choosiness favors the evolution of male choosiness, and that selection for mutual mate choice should be common. In turn, because female and male choosiness are based on the same phenotypic trait, male choosiness weakens indirect selection on female choosiness. In finite populations, this causes coevolutionary dynamics of "preference cycling", initiated by drift and completed by selection, with strong potential to destabilize reproductive isolation.

## Results

**Model overview.** We model the evolution of assortative mating in sympatry, based on three diploid biallelic loci that segregate independently (no physical linkage). Disruptive viability selection acts on an ecological locus $A$, but without mate choice, ecological divergence is hampered by random mating that brings divergent ecotypes (AA and aa) together to hybridize. In addition, we implement two distinct choosiness loci $F$ and $M$, which are independently expressed in females and in males, respectively. Both sexes can therefore use the trait under disruptive viability selection as a basis for mate choice (by using a matching rule). Female and male choosiness are ecologically neutral, but they can experience indirect selection via linkage disequilibrium with the ecological locus. Hybridization rates between ecotypes may decline due to assortative mating caused by female, male, or mutual preferences. Unless stated otherwise, we assume that the alleles coding for choosiness are recessive (only FF females and MM males are choosy).

Each generation first undergoes disruptive viability selection with strength $s$, such that heterozygotes at the ecological locus (Aa) suffer increased mortality. Males then court females and are "visible" to them (i.e., available as potential mates) proportionally to the courtship effort they invest. Choosy males (MM) prefer to court females that match their own ecological trait. In the case of a mismatch, they reduce their courtship effort to a very small fraction $\epsilon_m \ll 1$ of what nonchoosy males would invest. The courtship effort

thus saved can be reallocated toward preferred females, where the extent of this reallocation is described by the parameter $\alpha$. In particular, if choosy males reallocate all saved courtship effort toward preferred females ($\alpha = 1$), they enjoy a strong mating advantage over nonchoosy males with these particular females. Females likewise express different propensities to accept courting males. Choosy females (FF) prefer males that match their own ecological trait. We assume that in the case of a mismatch, they reduce the probability of mating to a very small value $\epsilon_f << 1$. Small $\epsilon_f$ and $\epsilon_m$ therefore reflect strong choosiness. Unlike males, all females have the same mating success as is the case in many polygynous mating systems. The expected genotype frequencies in the next generation depend on the probabilities of mating between different genotypes, with the new generation being formed by assuming independent Mendelian inheritance at all loci.

This three-locus diploid model of mutual mate choice is too complex to produce analytical solutions[34]. The behavior of the model can be assessed, however, by numerical analyses and computer simulations. We first analyze the deterministic behavior of the model, assuming an infinite population. Subsequently, we perform stochastic simulations in populations of finite, yet appreciable, size to account for genetic drift affecting traits under weak selection. In each generation, stochasticity is introduced by sampling $K$ offspring individuals following the distribution of genotype frequencies predicted by the deterministic model (just as in the Wright–Fisher model of genetic drift). In addition, in these stochastic simulations, we allow for mutation of alleles in offspring.

**Viability and sexual selection on female and male choosiness.** Viability and sexual selection may act directly on ecological or choosiness loci (through differential viability and male mating success among genotypes, respectively; Supplementary Fig. 1), and despite free recombination, also generate linkage disequilibrium between loci by creating statistical associations between alleles (in diploid individuals and in haploid gametes; Supplementary Figs. 2 and 3). In particular, the majority of choosy females and choosy males are homozygous at the ecological locus. In addition, choosy males often carry alleles coding for female choosiness (which are neutral during courtship). This linkage disequilibrium between choosiness loci arises because both choosy females and choosy males use the same ecological trait as the basis of mate choice, and therefore, tend to mate with each other. Consequently, selection that changes frequencies at a given locus will also change frequencies at other loci. As detailed below, such indirect selection plays a key role in the evolution of male and female choosiness.

To understand the intricate interplay of selective forces, we first consider cases where choosiness can evolve in only one sex. Disruptive viability selection directly acts on the ecological locus (black arrow, Fig. 1) with homozygotes having a high viability (hereafter, "homozygous" and "heterozygous" refer to the genotype at the ecological locus). In addition, female choosiness can induce positive frequency-dependent sexual selection on the ecological locus (green arrow, Fig. 1), such that homozygous males have the highest mating success. Consequently, in the case where only female choosiness can evolve, female choosiness is favored by indirect viability and sexual selection due to linkage disequilibrium with the ecological locus (Fig. 1a; Supplementary Fig. 4).

The situation is rather different if only male choosiness can evolve. Unlike female choosiness, male choosiness does not induce direct sexual selection on the ecological locus, because females in our idealized polygyny scenario do not differ in their mating success, even if some receive less courtship than others. However, male choosiness intensifies male–male competition for the preferred female type, which induces sexual selection on the male choosiness

locus itself (pink arrow, Fig. 1). Because the majority of choosy males are homozygous at the ecological locus (linkage disequilibrium), males courting "popular" homozygous females face strong competition for mating opportunities (Supplementary Fig. 1). Choosy homozygous males therefore place themselves in a disadvantageous competitive setting and have low mating success. In addition, if reallocation of courtship effort is only partial ($\alpha < 1$), choosy males lose courtship opportunities, which further lowers their mating success (blue arrow, Fig. 1). Consequently, in the case where only male choosiness can evolve, male choosiness is favored only if negative sexual selection is offset by indirect viability selection due to linkage disequilibrium with the ecological locus (Fig. 1b; Supplementary Fig. 4).

We now turn to our main case, where choosiness can evolve in both sexes. In addition to selection acting on female and male choosiness separately (Fig. 1a, b), choosiness in each sex now induces selective forces on choosiness in the opposite sex. First, because choosy females mainly reject nonchoosy (nonmatching) males, female choosiness directly increases the mating success of choosy males (red arrow, Fig. 1c). Second, sexual selection induced by female choosiness on the ecological locus also indirectly favors male choosiness (dashed green arrow, Fig. 1c). Finally, due to the linkage disequilibrium between choosiness loci, all selective forces acting on male choosiness also indirectly affect the evolution of female choosiness (dashed pink, red, and blue arrows in Fig. 1c).

**Coevolution of female and male choosiness.** To characterize the resulting coevolutionary dynamics of female and male choosiness, we measured the change in frequencies of choosy females and choosy males (1) over one generation (resulting from the combined viability and sexual selection; arrows in Fig. 2b–f) and (2) during mating and reproduction (capturing the consequences of sexual selection alone; red and blue colors in Fig. 2b–f). Interestingly, for $\alpha > 0$, the evolution of female choosiness changes the direction of sexual selection acting on male choosiness (change from dark to light blue as the frequency of choosy females increases, Fig. 2c–f). It can promote the evolution of male choosiness in situations where it would otherwise not evolve (Fig. 2c; Supplementary Fig. 4). Male choosiness can likewise change the direction of (indirect) sexual selection on female choosiness, but in an opposite direction (change from light to dark red as the frequency of choosy males increases, Fig. 2b–f); in particular, male choosiness clearly inhibits the evolution of female choosiness if viability selection is weak (Fig. 2e–f; Supplementary Fig. 4).

Indirect sexual selection on female choosiness might, in the simplest settings, reflect direct sexual selection on male choosiness, transmitted via linkage disequilibrium between the $M$ and $F$ loci. However, the causalities can be more complex than that, as evidenced by cases where male choosiness increases in frequency, while female choosiness decreases in frequency in the life cycle stage that involves mating and reproduction (Fig. 2d–f, colors indicate changes during that life cycle stage). The reason is that net indirect selection results from linkage disequilibrium between all three loci. Choosy males that also carry alleles for female choosiness (genotypes FF–MM) are particularly likely to be homozygous at the ecological locus (AA or aa), whereas other types of choosy males (genotypes ff–MM and Ff–MM) are somewhat more likely to be Aa heterozygotes (Supplementary Fig. 2). Under these particular conditions, choosy homozygous males (unlike their heterozygous competitors) suffer intensified male–male competition, which generates indirect selection against female choosiness via linkage disequilibrium. As a whole, sexual selection can therefore favor male choosiness and simultaneously inhibit female choosiness.

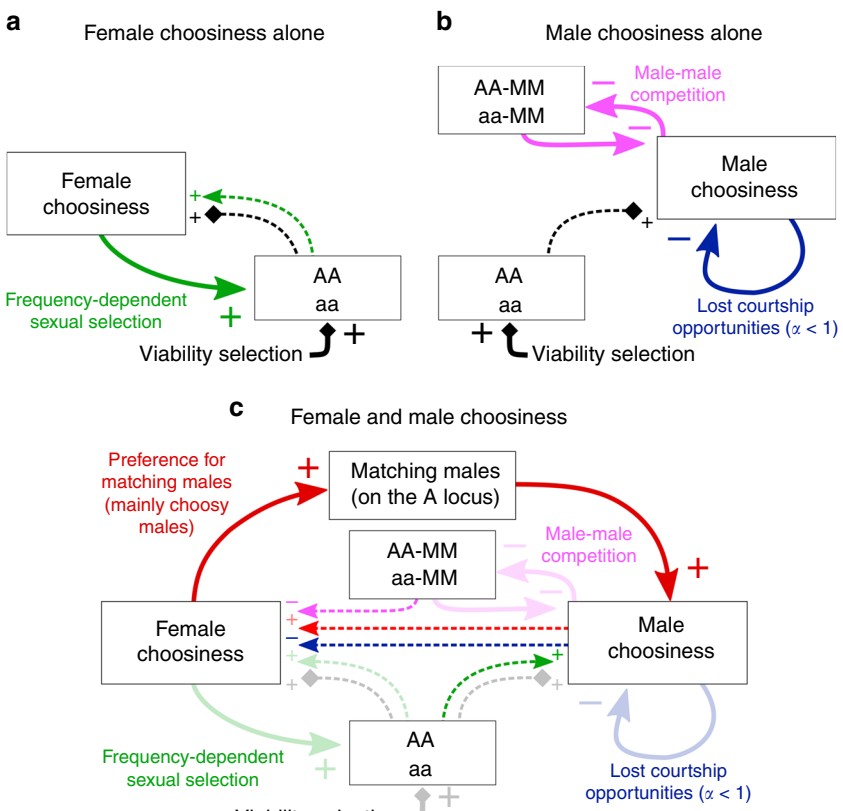

**Fig. 1** Selective forces acting on female and male choosiness. Choosiness can only evolve in one sex (**a**, **b**) or can evolve in both sexes (**c**). Diamond arrows and classic arrows represent viability and sexual selection, respectively. Viability selection acts through differential survival (i.e., change in frequencies during the disruptive viability selection process), whereas sexual selection acts through differential male mating success (detailed in Supplementary Fig. 1). Bold and dashed arrows represent direct and indirect selection, respectively. AA and aa refer to homozygotes at the ecological locus and MM to choosy males. Selective forces represented with a low opacity in panel **c** are the ones already shown in panels **a** and **b**

Finally, if males are choosy, females are at low risk of producing unfit hybrids, regardless of their own choosiness, as males focus their efforts on matching females. This weakens the linkage disequilibrium between the female choosiness locus and the ecological locus (Supplementary Fig. 2), and reduces indirect selection favoring female choosiness (cf. selection gradients in Fig. 4), with the consequence that female choosiness may easily drift in finite populations. The implication of such weak selection for choosiness coevolution will be assessed below by using stochastic simulations.

**Female and male choosiness in the absence of drift**. When viability selection is weak, four different choosiness regimes can evolve at deterministic equilibrium depending on the extent of reallocation of courtship effort ($s < 0.05$, Fig. 2a). In particular, if males do not reallocate courtship effort at all ($\alpha = 0$), assortative mating is based only on female choosiness, which is favored by indirect viability and sexual selection. On the contrary, if males fully reallocate courtship effort ($\alpha = 1$), sexual selection acting on male choosiness can indirectly inhibit the evolution of female choosiness (Fig. 2e–f); in this case, assortative mating is based only on complete or partial male choosiness (depending on the relative strength of viability and sexual selection; see Supplementary Fig. 4). Note that this outcome is not reached if females are initially choosy.

When viability selection is strong, mutual mate choice is a common deterministic equilibrium ($s > 0.15$, Fig. 2a). When females are nonchoosy, male choosiness only evolves if choosy males can reallocate most of their saved courtship effort (e.g., for

$\alpha \geq 0.8$ if $s = 0.2$, Supplementary Fig. 4). If female and male choosiness can coevolve, however, male choosiness evolves even if choosy males reallocate very little of their courtship effort toward preferred females (for $\alpha \geq 0.01$ if $s \geq 0.2$, Fig. 2a). As explained above, female choosiness favors male choosiness by changing the direction of sexual selection (Fig. 2c). Recall that while choosy individuals avoid courting/mating across ecotype boundaries, premating isolation is not complete ($\epsilon_m \neq 0$ and $\epsilon_f \neq 0$). Consequently, with mutual mate choice, female and male choosiness synergistically reduce hybridization rate between ecotypes (Supplementary Fig. 5). Finally, changing the dominance hierarchy (i.e., making alleles F or M dominant) does not change our results qualitatively (Supplementary Fig. 6).

**The consequences of drift on coevolutionary dynamics**. We next ran stochastic simulations to investigate the coevolutionary dynamics of male and female choosiness in populations of finite, yet appreciable, size ($K = 500$). Unless stated otherwise, we here consider scenarios with strong disruptive selection ($s = 0.2$), for which the deterministic outcome is female mate choice (for $\alpha \leq 0.01$) or mutual mate choice (for $\alpha > 0.01$). We define a frequency threshold ($= 0.85$) above which female or male populations are considered to be mainly choosy (i.e., mainly express a choosy behavior before mating). We thereby characterize four regimes of choosiness: female choice only ($\mathcal{F}$), male choice only ($\mathcal{M}$), mutual choice ($\mathcal{FM}$), and partial choice (i.e., both female and male populations are at most partly choosy, $\mathcal{P}$) (Fig. 3a). Note that regime $\mathcal{P}$ includes the regime of complete random

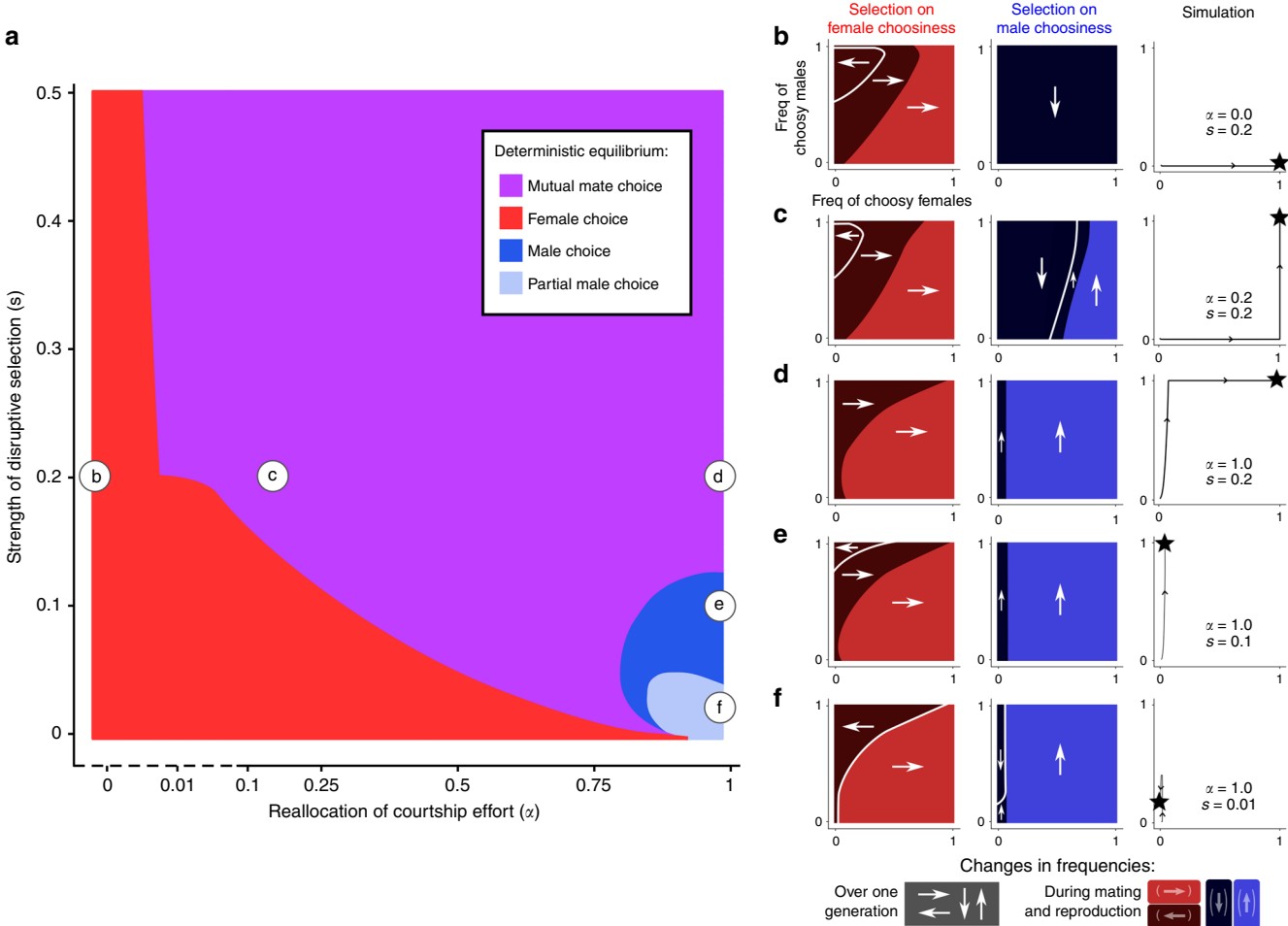

**Fig. 2** Deterministic equilibrium for different combinations of courtship reallocation ($\alpha$) and strength of disruptive viability selection ($s$). Panels **b**–**f** refer to specific combinations of ($\alpha, s$) marked in panel **a**. In the rightmost plot of panels **b**–**f**, we draw the trajectory of individual simulations leading to deterministic equilibrium (represented by black stars, and color-coded in panel **a**). We also measure the change in frequencies of choosy females and choosy males due to the combined action of all selective forces over one entire generation (white arrows). Indirect viability selection always increases the frequencies of choosy individuals (not shown). The change in frequencies of choosy females and choosy males during mating and reproduction (i.e., excluding the viability selection part of the life cycle) is represented by dark and light colors (dark: decrease in frequency, light: increase in frequency, as illustrated by the arrows in brackets in the legend). The colors in panels **b**–**f** thus capture the (direct and indirect) consequences of sexual selection. Depending on ($\alpha, s$), female mate choice, male mate choice, or mutual mate choice can be the stable deterministic equilibrium (**a**). In particular, under strong disruptive viability selection ($s \geq 0.2$), mutual mate choice evolves even if choosy males reallocate little courtship effort toward preferred females ($\alpha \geq 0.01$, note the logarithmic scale in the left part of the x-axis in panel **a**). In that case, the evolution of female choosiness favors the evolution of male choosiness by increasing the mating success of choosy males (leading to an increase in the frequency of choosy males during mating and reproduction in **c**)

mating and is therefore different from the deterministic equilibrium of partial male choice described above (Fig. 2a).

For $\alpha > 0.01$, our deterministic analysis predicts a stable equilibrium of mutual mate choice, yet with drift, choosiness traits can temporarily evolve away from this equilibrium (Fig. 3b), entering the regimes of male choice only (regime $\mathcal{M}$) or partial choice (regime $\mathcal{P}$). This is caused by drift-induced and selection-driven coevolutionary dynamics of female and male choosiness that we describe below.

Despite selection favoring mutual mate choice, assortative mating is often based solely on male choosiness in stochastic simulations (regime $\mathcal{M}$, Fig. 3b). When females are choosy, male choosiness is strongly favored, with drift playing an insignificant role. However, when males are choosy, selection favoring female choosiness is weak (as explained above); the frequency of choosy females may then decrease through drift (Fig. 4). Nonchoosy females can persist for significant periods of time, during which assortative mating is maintained by male choosiness only (regime $\mathcal{M}$).

Female and male populations are rarely simultaneously partly choosy (regime $\mathcal{P}$) for both low and high values of $\alpha$ (Fig. 3b). When $\alpha < 0.01$, male choosiness does not evolve and female choosiness is under sufficiently strong selection to remain at high frequency (regime $\mathcal{F}$). When $\alpha > 0.9$, choosy males can reallocate most of their courtship effort, and male choosiness is maintained by indirect viability selection even if female choosiness (and the associated sexual selection) is absent. The situation changes for intermediate values of $\alpha$, for which female and male populations are simultaneously partly choosy (regime $\mathcal{P}$) for significant periods of time (5% of time). The reason is that for intermediate $\alpha$, male choosiness is favored only when a large proportion of females are choosy. When the frequency of choosy females decreases due to drift, male choosiness becomes selected against, and the population can enter the regime of partial choosiness (regime $\mathcal{P}$, Fig. 4b–d). Although selection (first for female choosiness and then for male choosiness) will ultimately cause a return to mutual mate choice, this process takes time, and a snapshot of

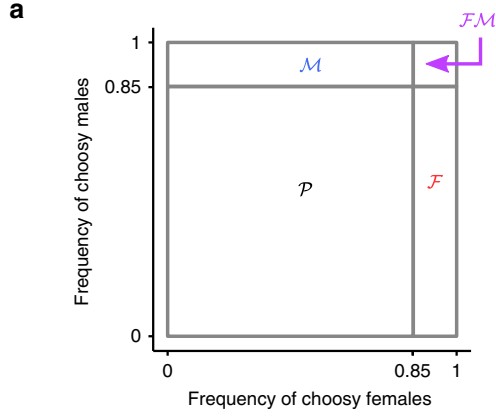

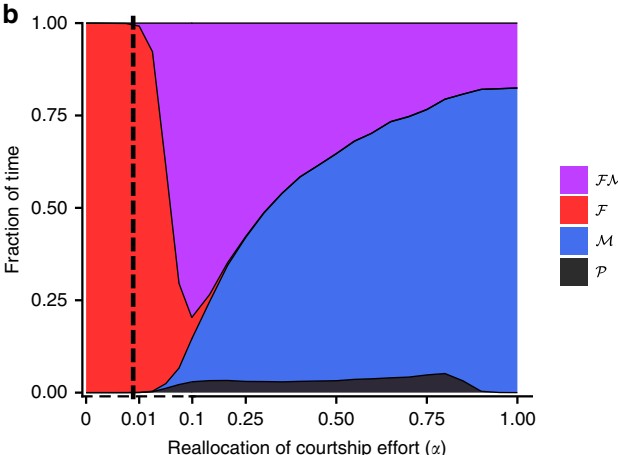

**Fig. 3** Time spent in each regime of choosiness in stochastic simulations. **a** Four regimes of choosiness are defined to describe stochastic simulations: mutual mate choice ($\mathcal{FM}$), male mate choice ($\mathcal{M}$), female mate choice ($\mathcal{F}$), and partial mate choice (i.e., both female and male choosiness occur at intermediate or low frequencies <0.85, $\mathcal{P}$). **b** Fractions of time spent in each regime as a function of the reallocation of courtship effort ($\alpha$), in stochastic simulations with $s = 0.2$ and $K = 500$ measured after the equilibrium predicted by the deterministic model has been reached for the first time. The vertical dashed line corresponds to the threshold above which the deterministic equilibrium is mutual mate choice ($\alpha > 0.01$, Fig. 2a). In stochastic simulations, in contrast, assortative mating is often based on male choosiness only (regime $\mathcal{M}$). Moreover, for partial reallocation of courtship effort ($\alpha \in [0.01, 0.9]$), female and male populations are partly choosy (regime $\mathcal{P}$) for significant time periods (5% of total time)

the population at a given point in time has a significant chance of observing the regime $\mathcal{P}$.

Hereafter, we refer to this coevolutionary dynamics as "preference cycling", since female and male choosiness go through deterministic cycles triggered by stochasticity, involving departure from regime $\mathcal{FM}$ into regimes $\mathcal{M}$, $\mathcal{P}$, and sometimes $\mathcal{F}$ (before returning to regime $\mathcal{FM}$). Preference cycling also occurs if we assume that females do not all have the same mating success (i.e., if we add a weak cost of female choosiness; Supplementary Note 1) or if choosiness is allowed to vary as a continuous trait (Supplementary Note 2).

**Preference cycling strongly increases hybridization rate.** Since we assume that neither sex can ever achieve perfect choosiness ($\epsilon_m \neq 0$ and $\epsilon_f \neq 0$), it is not surprising that hybridization rate is

the lowest in the regime of mutual mate choice ($\mathcal{FM}$) and becomes somewhat higher during drift-induced excursions into the male choice regime ($\mathcal{M}$) (blue area in Fig. 5d). More importantly, hybridization strongly increases during deterministic excursions into the partial choice regime ($\mathcal{P}$) (gray area in Fig. 5d). The greatest increase in hybridization occurs if the population stays in this regime $\mathcal{P}$ for extended periods of time. As shown in Fig. 5, while the regime $\mathcal{P}$ is reached most frequently for $\alpha \simeq 0.1$, the average time spent in this regime is maximal for $\alpha \simeq 0.8$, and during these episodes, maladapted heterozygotes reach frequencies of up to 35%. Overall, preference cycling leads to temporary peaks of hybridization, which periodically homogenize populations (as shown by fluctuations of $F_{ST}$ measured at neutral loci between genotypes AA and aa in Supplementary Fig. 7). In other words, even though mutual mate choice, whenever it occurs, achieves the strongest degree of assortative mating, it is also particularly prone to periodic breakdowns, which as a whole hamper the maintenance of premating isolation. Importantly, these periodic breakdowns of reproductive isolation do not occur if choosiness is allowed to evolve in one sex only (Supplementary Figs. 8 and 9).

If disruptive viability selection is weak (low $s$) or population size is small (low $K$), drift remains strong relative to indirect selection acting on female choosiness, and preference cycling occurs more frequently, increasing the overall hybridization rate (Supplementary Figs. 10 and 11). If alleles coding for choosiness are dominant instead of recessive, the final approach to fixation of the male choosiness allele is less rapid, and more nonchoosy males remain in the system. As a result, preference cycling induced by drift of female choosiness occurs rarely (Supplementary Fig. 12). Likewise, if male choosiness is less perfect, preference cycling occurs less frequently, and the overall hybridization rate is decreased (high $\epsilon_m = 0.03$ instead of 0.01 in Supplementary Figs. 13 and 14). This somewhat counterintuitive result is explained by the fact that with imperfect male choosiness, selection on female choosiness never becomes so weak as to be overwhelmed by drift. More generally, preference cycling may occur if the evolution of choosiness leads to nearly perfect reproductive isolation between ecotypes. This is the case if choosiness per se is nearly perfect (Supplementary Figs. 13 and 14), or if reproductive isolation is strengthened by additional barriers to gene flow (but reproductive isolation cannot drop below a certain value in that case; Supplementary Fig. 15). Under these conditions, which intuitively seem conducive to speciation, we show that coevolution of male and female choosiness has the potential to strongly destabilize reproductive isolation.

## Discussion

Surprising coevolutionary dynamics of male and female mate choice occur when the preferences of both sexes are based on the same phenotypic trait under disruptive selection. We showed that choosiness (i.e., the strength of preference) in one sex influences the evolution of choosiness in the other, a factor that has not been considered in previous models of speciation with gene flow[17–22,31,32]. Based on the predictions of our model, genetic drift, incomplete reallocation of courtship effort, and strength of choosiness all prove to be important when examining the outcome of this coevolutionary dynamics in terms of reproductive isolation.

In our model, male and female preferences are based on the same phenotypic trait, but are themselves governed by different loci. This genetic basis gives scope for preference coevolution. Selection generates linkage disequilibrium between choosiness and ecological loci (despite free recombination), and indirect viability and sexual selection resulting from this linkage disequilibrium has profound consequences for the evolution of

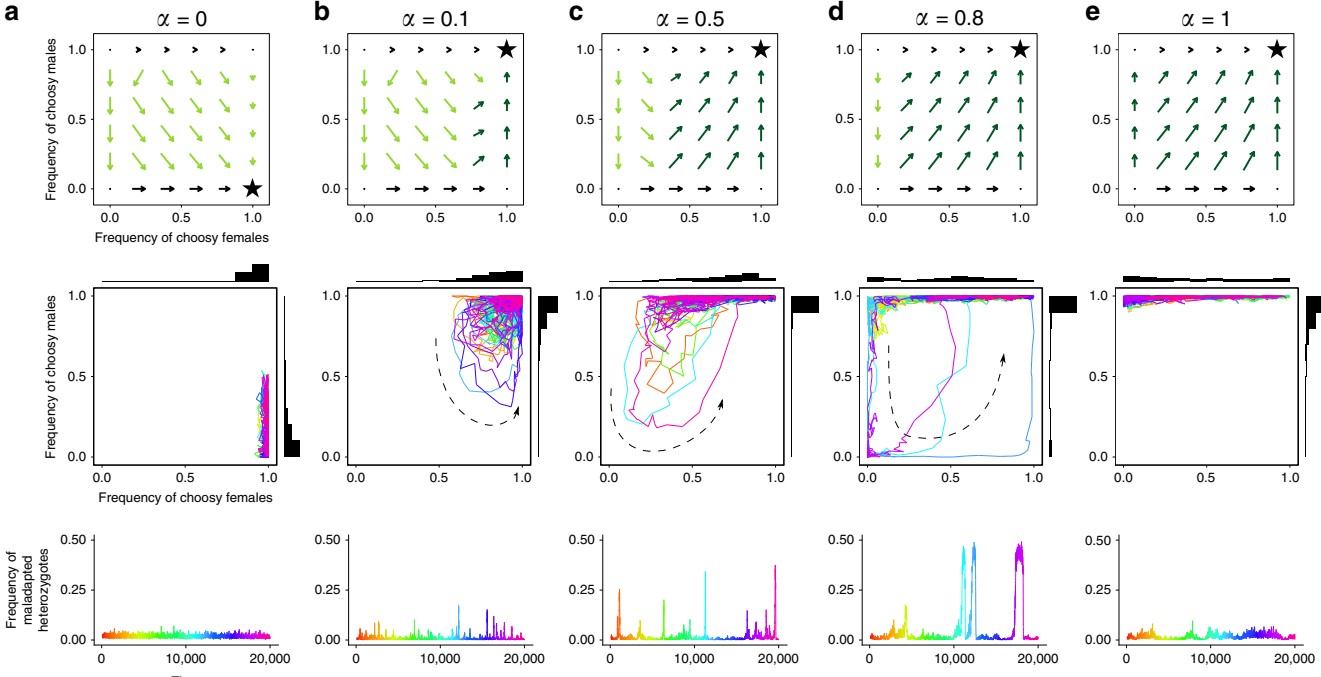

**Fig. 4** Selection gradients for female and male choosiness and examples of stochastic simulations ($s = 0.2$, $K = 500$) with different courtship reallocation values ($\alpha$). From top to bottom, we represent the deterministic selection gradients for female and male choosiness, the evolutionary dynamics of choosiness, and the resulting frequencies of maladapted heterozygotes at the ecological locus before viability selection (from the same simulations). The selection gradients for choosiness correspond to the relative change in frequencies of choosy versus nonchoosy females and males over one entire generation (net effect of viability and sexual selection combined) in the deterministic model, and are therefore simpler than the representation in Fig. 2b–f. To highlight weak selection on choosiness, the length of arrow vectors is drawn on a logarithmic scale. Stars correspond to the regimes of choosiness at deterministic equilibrium. The graphs in the second and third rows show stochastic simulations, started at the choosiness regime predicted by the deterministic analysis, and rainbow color gradients correspond to the passage of time. Note that in **b**–**d**, the stochastic simulations do not converge to the deterministic equilibrium, but instead display "preference cycling", the direction of which is represented by a dashed arrow

choosiness in both sexes. In particular, female and male mate choice can substitute for each other as drivers of assortative mating, but have different consequences on selection experienced by the opposite sex. Male choosiness relaxes indirect selection on female choosiness, because even nonchoosy females can avoid producing unfit hybrids when males focus their efforts on females of their own ecotype. While female choosiness reduces indirect selection for male choosiness in a similar manner, it also strongly favors male choosiness through direct sexual selection. This is because male choosiness is a poor strategy when females are not choosy, as choosy males focusing on a subset of females place themselves in a disadvantageous competitive setting[31] and may incur opportunity costs. If females are choosy, however, choosy males gain a high mating success by disproportionately courting those females that are likely to accept them.

Female choosiness therefore favors the evolution of male choosiness if at least some of the courtship effort saved by refraining from courting unpreferred females can be reallocated to gain a mating advantage with preferred females. As a consequence, male choosiness evolves more easily in our model than in models investigating the evolution of choosiness in each sex separately[31]. Indeed, our results show that mutual mate choice should often be favored by selection, and can induce very strong reproductive isolation in infinite populations.

Counterintuitively, however, in finite populations, this regime of mutual mate choice is particularly unstable. In the presence of even weak genetic drift, the coevolutionary dynamics of female and male mate preferences can lead to transient but periodic breakdowns of premating isolation, strongly increasing the hybridization rate. The fact that either sex can cause

assortative mating makes it difficult for mutual mate choice to be maintained; more precisely, if male preferences are sufficiently strong to establish assortativeness, female choosiness has little effect on the mating outcome and is therefore free to drift. When female choosiness is reduced, male choosiness becomes disfavored by selection, temporarily leading to a regime of random mating. This coevolutionary dynamics of "preference cycling", initiated by drift and completed by selection, strongly destabilizes reproductive isolation and leads to periods of increased hybridization, which homogenizes populations.

The establishment of premating isolation is often considered to be the first step toward speciation with gene flow, and its stability is therefore a key prerequisite for other isolating barriers to evolve. For instance, only stable premating isolation should allow the establishment of neutral genetic incompatibilities leading to subsequent postzygotic isolation[55–57]. If selection favoring mutual mate choice induces preference cycling and dynamic instability of premating isolation, our results suggest that it can become an obstacle to speciation. This scenario contrasts with the traditional view of speciation as a gradual process characterized by a constant accumulation of barriers to gene flow (so-called "speciation continuum")[2,58–60]. Speciation can also be "undone"; like assortative mating in our model, barriers to gene flow can dissolve, and genetic discontinuities may vanish, thereby merging two taxa into a single population by hybridization[61,62]. Our model predicts such cycles of divergence, and gene flow may actually characterize the process of diversification in nature.

To track the "speciation continuum", empirical research often estimates isolating barriers between pairs of populations varying in their level of differentiation[2,58]. Yet, it is important to

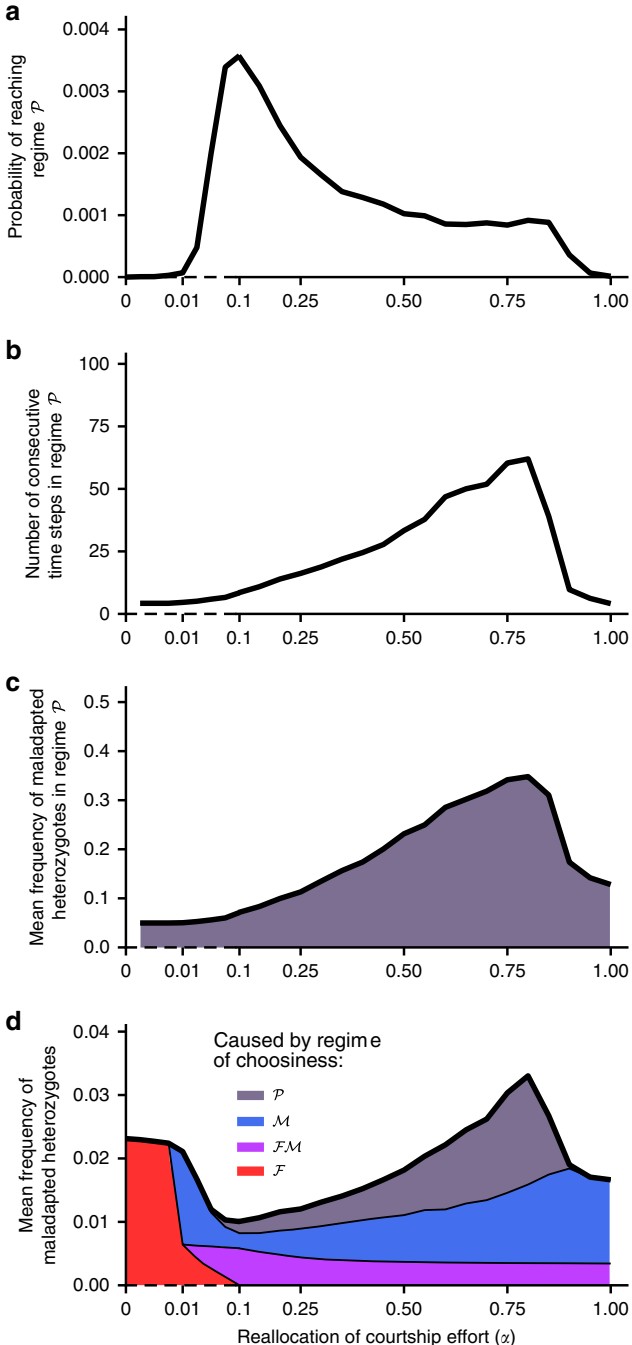

**Fig. 5** Coevolutionary dynamics of choosiness ("preference cycling") and the resulting hybridization rate in stochastic simulations ($s = 0.2$, $K = 500$). To describe the coevolutionary dynamics of female and male choosiness, we record the mean probability of reaching regime $\mathcal{P}$ from regime $\mathcal{FM}$ or $\mathcal{M}$ at each time step (**a**), the mean number of consecutive time steps in regime $\mathcal{P}$ (**b**), and the mean frequency of maladapted heterozygotes (Aa at locus $A$) in regime $\mathcal{P}$ before viability selection (**c**) as a function of the reallocation of courtship effort ($\alpha$). To assess the resulting hybridization rate, we record the mean frequency of maladapted heterozygotes over evolutionary time (**d**). In panel **d**, we also represent the mean contribution of each regime of choosiness to hybridization. In stochastic simulations, hybridization rate is increased by the coevolutionary dynamics of female and male choosiness despite selection favoring mutual mate choice (when $\alpha > 0.01$) (**d**). In particular, the coevolution of female and male choosiness leads to periodic episodes of random mating, strongly increasing the hybridization rate (with up to 35% of hybridization in regime $\mathcal{P}$) (**c**)

on how much courtship effort males can reallocate toward preferred females, as a result of foregoing courting unpreferred females. By treating the reallocation of courtship effort as a parameter ($\alpha$), our model covers a wide variety of courtship and mating systems in animals. In particular, "courtship effort" may refer to time (e.g., for mate searching or performing complex displays[38]) or energy (e.g., for resource-demanding spermatophores[36] or nuptial gifts[37]). In addition, foregoing certain courtship opportunities and searching for more preferred mates might entail mortality costs, which also affect the extent of reallocation of courtship effort. Our results suggest that obtaining estimates of courtship reallocation in nature is important to increase our understanding of divergence in the presence of gene flow. Based on our predictions, selection should favor mutual mate choice with even little reallocation; without reallocation, in contrast, male choice should be deleterious. Importantly, if reallocation is partial, preference cycling may occur, possibly limiting divergence.

More generally, while female choosiness is abundant in nature, male choosiness usually associates with particular mating systems (e.g., monogamy) and often evolves in addition to female choosiness[42–45] (in agreement with our predictions). Yet, we showed that the evolution of male choosiness can destabilize reproductive isolation. Therefore, the mating system characterizing each taxon should determine the stability of reproductive isolation and therefore the likelihood of speciation. Further empirical studies could usefully add this consideration when testing for links between the mating system and speciation; our model suggests that taxa with mating systems that are prone to the evolution of male mate choice may not necessarily associate with high speciation rates.

The extent and stability of reproductive isolation also depends on how accurately existing preferences can be expressed. Imperfect preference may occur for many reasons. For instance, in the butterflies *Heliconius melpomene* and *Heliconius cydno*, female and male preference loci are physically linked with different color pattern loci[50]. Therefore, choosy individuals may not completely stop courting/mating across ecotype boundaries because each sex relies on different aspects of the phenotype. Our model predicts that this error-proneness could strengthen selection favoring mutual mate choice, which could in turn inhibit preference cycling and stabilize reproductive isolation. Thus, perhaps counterintuitively, our model suggests that imperfect male preferences lead to strong reproductive isolation in the long-term by maintaining selection favoring female preferences, and preventing drift-induced preference cycling. Such counterintuitive results are not unheard of: in the context of local adaptation, imperfect female choice has been shown to be more strongly favored by selection than perfect choice because it

remember that such measures, being snapshots in time, do not yield information on the long-term stability of reproductive isolation. Our predictions regarding the coevolutionary dynamics of male and female mate preferences mean that premating isolation caused by mutual mate choice should be interpreted cautiously: whether gene flow is reduced over long periods of time is an open question. If a species' range is partially fragmented, preference cycling could also cause variation in the degree of reproductive isolation among local populations, as observed, for instance, in *Catostomus* fish species[63,64]. Yet, variation in hybridization is rarely quantified across several natural populations, and our study highlights the value of studies characterizing the strength of isolating barriers at a broader spatial and temporal scale.

The coevolutionary dynamics of female and male mate preferences (and the resulting reproductive isolation) depends crucially

maintains a higher diversity of male types in the population[65]. Likewise, many theoretical studies have found cases where sexual selection favors partial choosiness[16,24–26]. Our study adds to this quest the possibility of preference cycling in situations where choosiness evolves as a quantitative trait (see Supplementary Note 2). In particular, if choosiness only evolves to a partial degree, preference cycling may not occur; interestingly, this may favor speciation.

Our predictions are not limited to the context of emerging reproductive isolation among diverging populations in sympatry. We can expect similar coevolutionary dynamics of female and male preferences in more advanced stages of reproductive isolation, e.g., after secondary contact. Indeed, disruptive viability selection may be caused by genetic incompatibilities among more distantly related taxa. In this context, preference cycling could likewise temporally increase hybridization rate, and conceivably, explain the formation of "hybrid swarms" and subsequent genetic introgression[5] or hybrid speciation[66].

Overall, our theoretical model adds support to the idea that premating isolation may often be readily reversible[27,56,67]. Intriguingly, we show that premating isolation should be particularly unstable under conditions that favor mutual mate choice. We highlighted some factors that could inhibit preference cycling (strong selection against hybrids, high carrying capacity, imperfect choices, and extensive reallocation of courtship effort). The geographical context of speciation and a more detailed look into alternative genetic architectures (e.g., "two-allele mechanisms"[68], physical linkage among choosiness loci) could conceivably change the modalities of preference cycling and should therefore be investigated in future theoretical studies. It will also be fruitful to consider preference cycling in a system with multiple potential barriers to gene flow, for if the time between episodes of hybridization during preference cycling is very long, other barriers might have time to evolve despite preference coevolution taking place. On the empirical side, the occurrence of preference cycling and its impact on reproductive isolation remains to be tested. More generally, our study should stimulate further research on the stability of barriers to gene flow, and on the link between mating systems and speciation.

Assuming discrete generations, we follow the evolution of genotype frequencies $\mathbf{p}(t)$ within an infinite population (deterministic simulations) or within a finite population (stochastic simulations). $\mathbf{p} = \{p_i\}$ is a vector consisting of 27 elements $\{p_1, p_2, ..., p_{27}\}$ referring to the frequencies of the 27 genotypes present in newborn offspring. The life cycle is as follow: census, viability selection, courtship/mating, zygote formation (and random sampling in stochastic simulations).

**Disruptive viability selection on the ecological locus.** Environmental/ecological pressures act on an adaptive ecological trait and favour sympatric divergence into two distinct ecotypes occupying niches of equal size. To ensure the maintenance of polymorphism, a parameter $s' > 0$ confers an advantage to the rarer of the two homozygotes AA and aa. We also assume that heterozygotes Aa suffer viability costs described by a parameter $s$. Following these assumptions, the genotype frequencies after viability selection are:

$$p_i^S = \begin{cases} p_i\left(1 + s'\left(0.5 - \frac{\sum_{k \in aa} p_k}{\sum_{k \in aa \cup AA} p_k}\right)\right)/\hat{N}, & \text{if } i \in aa \\ p_i\left(1 + s'\left(0.5 - \frac{\sum_{k \in AA} p_k}{\sum_{k \in aa \cup AA} p_k}\right)\right)/\hat{N}, & \text{if } i \in AA \\ p_i(1-s)/\hat{N}, & \text{if } i \in Aa \end{cases} \quad (1)$$

The normalization factor $\hat{N}$ ensures that the genotype frequencies $p_i^S$ sum up to 1. If $s' = 0$, the ecological allele that is more frequent initially may outcompete the other allele ("gene swamping"[69]), hampering divergence[23,70,71]. To prevent fixation of a single ecological genotype, we always include some negative frequency dependence by implementing $s' > 0$ (leading to $p_{AA}^S \approx p_{aa}^S$).

**Male choice and courtship.** $P_{m,f}^{\male}$ denotes the courtship effort of a male with genotype $m$ towards females with genotype $f$ ($m$ and $f \in \{1, 2, ..., 27\}$). Males with genotype mm or Mm at locus $M$ are nonchoosy and court all females with the same intensity ($P_{m,f}^{\male} = 1$ toward all females). Homozygous MM males are choosy (i.e., they express homotypic preferences), and their courtship depends on the match between the ecological trait (locus $A$) of the female and their own. In case of a mismatch (e.g., between a male with genotype AA and a female with genotype Aa or aa), choosy males reduce their courtship effort to a small fraction $P_{m,f}^{\male} = \epsilon_m << 1$ (small $\epsilon_m$ thus reflects strong choosiness). In other words, choosy males reduce resources (e.g., time or energy) spent on courting unpreferred females. Saved courtship effort can be reallocated (totally, partially, or not at all) toward courtship of preferred (matching) females. The extent of this reallocation is described by parameter $\alpha$. Overall, of all possible courtship events that could happen in the population, a fraction $C_{mf}$ will occur between males of genotype $m$ and females of genotype $f$:

$$C_{m,f} = p_m^S \, p_f^S \left(\overbrace{P_{m,f}^{\male}}^{\substack{\text{Baseline courtship effort of} \\ \text{a male of genotype } m \text{ towards} \\ \text{a female of genotype } f}} + \overbrace{\alpha\left(1 - \sum_{f'=1}^{27} p_{f'}^S P_{m,f'}^{\male}\right)}^{\substack{\text{Courtship effort that} \\ \text{a male of genotype } m \\ \text{reallocates}}} \times \overbrace{\frac{P_{m,f}^{\male}}{\sum_{f'=1}^{27} p_{f'}^S P_{m,f'}^{\male}}}^{\substack{\text{Proportion of courtship effort that} \\ \text{a male of genotype } m \text{ reallocates} \\ \text{towards a female of genotype } f}}\right) \quad (2)$$

$\overbrace{\phantom{xxxxxxxxxxxxxxxxxxxxxxxxxxxxxxxxxxxxxxxxxxx}}^{\text{Total courtship effort of a male of genotype } m \text{ towards a female of genotype } f}$

## Methods

**Genotypes.** Our population genetic model is based on three autosomal diploid loci. Alternative alleles at each locus are represented by small and capital letters. An ecological locus, $A$, is subject to disruptive selection and can be used as a basis for mate choice (so called "magic trait"). Additional loci $F$ and $M$ alter female and male choosiness (i.e., strengths of homotypic preference) before mating. We assume that choosiness alleles code for either no choosiness or strong choosiness, i.e., preferences vary from indiscriminate to almost fully assortative. We assume no physical linkage (i.e., loci are on different chromosomes or very far apart on the same chromosome) and alleles assort independently of one another in gametes following Mendel's second law. This assumption allows us to reduce the number of dynamic variables needed to describe our genetic system. There are three genotypes per locus (e.g., AA, Aa, and aa for the $A$ locus), and, therefore, we track the frequencies of $3^3 = 27$ genotypes in the population.

where $p_m^S$ and $p_f^S$ are the frequencies of males of genotype $m$ and females of genotype $f$ after viability selection. If $\alpha = 1$, choosy males reallocate all saved courtship effort toward preferred females and therefore enjoy a strong mating advantage over their competitors with these particular females. Contrary to previous models[31,34,39,72], however, male preferences can induce lost courtship opportunities. If $\alpha < 1$, only part of the saved courtship effort is reallocated ($\sum_m \sum_f C_{mf} < 1$), and total courtship effort may differ between individual males ($\sum_f C_{mf} \neq p_m^S$). Equation (2) therefore differs from those previous models and is instead analogous to a model of female mating preferences with opportunity costs[25].

**Female choice and mating.** We assume that males are "visible" to females (i.e., available as potential mates) proportionally to their courtship effort, defining a baseline mating rate which can then be adjusted downwards or upwards by female

choice. $P^{\female}_{f,m}$ denotes the willingness of a female with genotype $f$ to mate with males with genotype $m$. Females with genotype ff or Ff at the locus $F$ mate indiscriminately ($P^{\female}_{f,m} = 1$ with all males), leading to mating rates that are directly proportional to courtship efforts. Homozygous FF females are choosy (i.e., they express homotypic preferences). Their decision to mate depends on the match between the ecological trait of the male and their own. In case of a mismatch, choosy females reduce the probability of mating to a small fraction $P^{\female}_{f,m} = \epsilon_f << 1$ of the baseline (small $\epsilon_f$ reflects strong choosiness). Thus, the overall proportion of matings $M_{mf}$ that occur between males of genotype $m$ and females of genotype $f$ is given by

$$M_{m,f} = \frac{C_{m,f} P^{\female}_{f,m}}{\sum_{m'=1}^{27} C_{m',f} P^{\female}_{f,m'}} \times p^{S}_{f} \qquad (3)$$

This equation is analogous to previous population genetic models of mating with female preferences[16,31,33,34,39,72]. It ensures that all females, even the ones that are less preferred by males (or that prefer rare males), have the same mating success (no cost of choosiness and no sexual selection from male choice). Likewise, the mating success of females with and without preference is identical ($\sum_m M_{mf} = p^{S}_{f}$). These assumptions are realistic for many polygynous mating systems; relaxing them by implementing a weak cost of female choosiness ($\sum_m M_{mf} < p^{S}_{f}$ for $f \in \{ \mathrm{FF} \}$) does not change our conclusions (whereas female choosiness does not evolve if it associates with a strong cost; see Supplementary Note 1).

**Zygote formation.** Expected genotype frequencies $\mathbf{p}(t+1)$ of zygotes in the next generation are calculated by summing the appropriate mating frequencies $M_{mf}$, assuming Mendelian segregation and free recombination between all loci.

**Random sampling for stochastic simulations.** Based on the above deterministic model, we also perform stochastic simulations in finite populations to account for drift at loci under weak selection. To do so, for each generation, we first apply Eqs (1) to (3) to the vector $\mathbf{p}(t)$, yielding the expected frequency distribution of genotypes in generation $t + 1$. The new vector $\mathbf{p}(t + 1)$ is then obtained by randomly sampling $K$ offspring individuals from this distribution. In addition, we assume that mutation can occur in each offspring individual with a probability $\mu$ per diploid locus.

**Parameters and initialization.** Unless stated otherwise, we perform simulations with strong choosiness in genotypes FF and MM ($\epsilon_f = 0.01$, $\epsilon_m = 0.01$). We also set $s' = 0.5$ to ensure that polymorphism at the ecological locus $A$ is maintained. Simulations start with only alleles f and m present in the population. Once the population has reached ecological equilibrium, 1% of choosy males and females are introduced at Hardy–Weinberg equilibrium, such that choosiness alleles are in linkage equilibrium with each other and with alleles at the ecological locus.

Deterministic equilibrium is typically reached in <1000 generations. In stochastic simulations, we model populations of appreciable size $K = 500$ with a probability of mutation $\mu = 10^{-3}$ per individual and per diploid locus. For each parameter combination, 40 stochastic simulations were run for 100,000 generations and statistics were calculated by averaging over time within run and over these runs.

Note that in previous models[23–26], sexual selection created by female choosiness has been shown to impede speciation if mating is initially random (such that intermediate phenotypes have the highest mating success) and if choosiness evolves in small steps. This is not the case in our model, where alleles conferring strong choosiness are allowed to directly compete with alleles for random mating (Supplementary Fig. 4). This allows a strong linkage disequilibrium to develop between FF genotypes and AA or aa genotypes, such that choosy females are mostly homozygous at the ecological locus, increasing the mating success of homozygous males and indirectly favouring ecological divergence.

**Reporting summary.** Further information on research design is available in the Nature Research Reporting Summary linked to this article.

## Data availability

All relevant data are available from the authors.

## Code availability

All simulations were run using Julia (version 1.0.1). The computer code of the simulations and of the analyses is provided as Supplementary Software.

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

## Acknowledgements

We thank O. Cotto and V. Llaurens for discussion and suggestions on the manuscript.
T.G.A. was funded by a PhD scholarship from the French Ministry of Higher Education and Research. This research was supported by grants from the Doctoral School GAIA (to T.G.A.), the French National Agency for Research (ANR-12-JSV7-0005-01-Hybevol and ANR-18-CE02-0019-01-Supergene) (to M.J.) and the Swiss National Science Foundation (to H.K.).

## Author contributions

T.G.A., M.J. and H.K. conceived the study, T.G.A. performed modeling work, T.G.A., M.J. and H.K. contributed to writing the paper.

## Competing interests
The authors declare no competing interests.
