## [Peer Review File · Nature Communications]

Reviewers' Comments:

Reviewer #1:

Remarks to the Author:

The authors analyze a three-locus, two-allele model of sympatric speciation by assortative mating due to mutual mate choice. They consider an ecological locus A under frequency-dependent disruptive selection, such that heterozygotes have a fitness disadvantage. This same trait also serves as the basis of mate choice ("magic trait") for both sexes (mutual choice) via a matching rule (homotypic preference, i.e., individuals prefer partners that have the same trait value as themselves). Individuals are either (almost perfectly) choosy or non-choosy, depending on their genotype at choosiness loci M for males and F for females (with the choosiness allele being recessive). Mating choice involves two stages: males court females, and females accept/reject courting males. Choosy males court mostly matching females that resemble themselves with respect to the ecological trait, and choosy females accept mostly matching males. Females are guaranteed to mate (no cost of choosiness, but this assumption is relaxed in an Appendix), whereas males may mate several times or not at all (strict polygyny). An key parameter is the proportion of courtship "effort" that choosy males can reallocate from unpreferred to preferred females. If this parameter is less than one, choosy males lose courtship opportunities.

The authors give a detailed account of the various direct and indirect selection pressures acting on female and male choosiness. In short, choosiness in both sexes is favored by indirect viability selection, since choosy individuals produce few heterozygote offspring (this is the classical reinforcement mechanism, which also operates in many models of sympatric speciation). In addition, male choosiness is under direct sexual selection, whose direction depends on female choosiness: if females are not choosy, males should not be choosy, either (from a sexual selection perspective), because choosy males lose courtship opportunities (if reallocation is imperfect) and place themselves into competitive situations with other choosy males. If females are choosy, however, males should be choosy, too, to avoid being rejected when courting non-matching females. Finally, both male and female choosiness are under various indirect selection pressures due to linkage disequilibrium between all three loci.

In this setting, mutual mate choice is frequently favored by selection: Basically, females (and males) should be choosy to prevent hybrid offspring, and males should be choosy to avoid being rejected by choosy females (provided reallocation of courtship effort is sufficiently large). However, the key effect in the model is that, once all males are (highly) choosy, selection on female choosiness vanishes (since females are almost only courted by matching males anyway, so they have nothing to choose from). In a finite population, female choosiness can therefore degrade due to mutation and genetic drift. If courtship allocation is high the population may then stay in a regime where assortative mating is driven by male choosiness only. If courtship allocation is low, however, strongly reduced female choosiness reverses selection on male choosiness, which will subsequently decline, too, leading to a regime of partial choosiness, which is close to random mating. Finally, reduced male choosiness re-establishes selection for female choosiness, which will increase again, dragging male choosiness with it. This pattern of "preference cycling" (triggered by drift, but completed by selection), which induces periodic episodes of hybridization, is the key result of this paper. Basically, the fact that assortative mating can be assured by both sexes makes it more difficult to maintain in the long term.

Preference cycling is a novel and interesting phenomenon that is principally suited for publication in Nature Communications. However, the paper will need serious revision before it can be accepted. My major comments are the following:

- 1) The writing needs serious improvement. I have extracted the main text from the pdf and attach a

.doc-file with (lots of) suggestions for revision (as well as comments; I didn't have time to do the same for the Appendices and supplementary Figures). I apologize to the authors if this comes across as condescending, which is not my intention. Of course, these suggestions are not obligatory.

2a) A potential limitation of the results is that 'preference cycling' only occurs if (male?) choosiness is nearly perfect. According to Figure S8, preference cycling vanishes if $\epsilon = 0.03$, that is, if choosy males have a three percent probability of courting a non-matching female. In this case, female choosiness remains under selection and does not degrade due to drift. One might wonder if $\epsilon = 0.01$ (as in most simulations) is realistic in nature, unless subpopulations are already strongly diverged. At the very least, this requires some discussion.

2b) Regarding the last point, I wonder if preference cycling might be more prevalent if female choosiness is costly (the authors briefly discuss costs for females in Appendix A, and conclude that the results are not qualitatively altered if females can reallocate at least 85% of rejected mating opportunities to preferred males). In this case, female choosiness might become neutral or deleterious already at lower values of male choosiness. Indeed, if choosiness is costly for both sexes, an almost game-theoretical situation arises, where both sexes benefit from assortative mating in terms of offspring fitness, but would prefer it to be taken care of by the other sex (not expressing choosiness would then be akin to cheating). Whether or not such a scenario arises might depend on the exact form of costs.

3) One issue I do not understand is negative indirect sexual selection on female choosiness as shown in Figure 2 ("sex-"), which is responsible for the male-choice and partial male-choice regimes in the deterministic model. Indirect negative sexual selection on female choosiness means that choosy females will have choosy sons (due to LD between M and F), whose mating success will be low (because of lost mating opportunities and high competition). This should happen if there are too many choosy males relative to the number of choosy females. However, there is one thing I don't understand at all: How is it possible that there are "sex-" regions for females that overlap with "sex+" regions for males (Fig. 2c-e)? Since sexual selection on females can only be indirect (as all females have by definition equal mating success), how can it be negative on females when it is positive on males? This needs to be clarified. Also, explain in a bit more detail what happens in Figures 2d and 2e (see my comments in the attached file).

4) Model with continuous choosiness (Appendix B): Are the low but positive fixation probabilities in Fig. B1 due to selection or drift? I suppose you need to take into account chance fixation of neutral or deleterious alleles in order to model genetic drift? Most importantly, what is happening with male choosiness for $\alpha = 0$? Why does it fluctuate quasi-periodically? Shouldn't it be selected against?

5) You say that you always use values of s (strength of disruptive selection) that assure that homozygotes are more frequent than heterozygotes, such that heterozygotes are never favored by sexual selection. However, this cannot be true in, e.g., Figure 2, where you explore all values of s between 0 and 1. Does sexual selection for heterozygotes play a role in these cases?

6) Can you say a bit more about possible effect of your choice of recessive choosiness alleles? For example, this should affect the fixation probabilities in Appendix B.

7) In the introduction and/or discussion, say a bit more about the results of previous models of speciation with male or mutual mate choice (e.g., ref. 38).

8) The effects of preference cycling on speciation will depend on its timescale (e.g., it is most severe when the population stays in the P regime for a long time). Can you say a bit more about this? How

long does one cycle take, i.e., what is the mean time between episodes of hybridization? Is it so long that other isolating barriers might conceivably already have evolved?

9) What do you think happens if loci are more tightly linked ($r < 0.5$)?

Signed,

Michael Kopp.

Reviewer #2:

Remarks to the Author:

In this paper, the authors investigate sexual selection by mutual mate choice using a mathematical model. The modeling and analysis seem sound and the results are interesting. However, the analysis does not correspond to the argument, which decreases the value of the paper. If the authors add some analyses to confirm the argument, the paper will be more attractive and acceptable to Nature Communications.

Major comment:

In the stochastic model, the frequency of heterozygotes (hybridization rate) increases under some conditions, which, the authors argue, is caused by mutual mate choice. However, if the preference loci are absent (no mate choice), the hybridization rate should be almost 0.5 before selection, which is clearly higher than that under mutual mate choice. Therefore, some presentations (e.g., title: Coevolution of male and female mate choice can destabilize reproductive isolation) are misleading because assortative mate choice necessarily decreases the hybridization rate. I guess that the authors want to argue that mutual mate choice is less effective for reproductive isolation than mate choice by single sex. If so, they should compare the hybridization rate under mutual mate choice with those under single mate choices. It is easy to make the model simulation as female choosiness alone and male choosiness alone (keep the frequency of M or F as zero). Without the comparisons, their argument is never confirmed.

Reviewer #3:

Remarks to the Author:

This paper presents results from a simulation study of the evolution of assortative mating. It concludes that isolation between species caused by assortment may be less robust than previously thought. Unfortunately, the model has a fundamental flaw in its mathematical foundation that requires the work be entirely redone. I also have reservations about the scope of the model's applications. I therefore do not recommend publication.

The foundational problem involves the genetic bookkeeping. The model tracks the frequencies of the three genotypes at each of three loci, for a total of $3 \times 3 = 27$ dynamic variables (l. 117-120). But in fact this genetic system has 36 dynamic variables. Take the simpler case of just two loci, A and B. The frequencies we need to follow are those of all the diploid genotypes, keeping track of which alleles were inherited together from the same gamete:

ab/ab, ab/aB, ab/Ab, ab/AB,
aB/aB, aB/Ab, aB/aB,
Ab/Ab, Ab/AB,

AB/AB

Thus there are 10 frequencies, rather than the $3^2 = 9$ frequencies that the ms would claim. In general, with n diallelic loci, there are 2^n types of gametes. The total number of diploid genotypes is

$$2^n + (2^n - 1) + \dots = 2^{n-1} (2^n + 1) ,$$

not 3^n as claimed in the ms.

Thus it appears that the recursion equations that are the basis for this study are wrong. It is possible that doing the model correctly will lead to similar conclusions, but that is unknown. I believe that the only case in which the ms' equations predict the dynamic behaviour correctly is when there is no linkage disequilibrium. The nonrandom mating in this model would seem to generate linkage disequilibrium, and so I suspect the dynamics are wrong.

A second consideration makes me feel that this paper would not be a good fit for Nature Communications even if the math was right. The model is not biologically general: the mating rules, number of loci, dominance relations, and lack of pleiotropy in the behaviour of males and females are just some of the limitations. Unfortunately, the paper includes only simulations -- general analytic results are entirely lacking. So it is not clear which results are likely to be biologically robust, and which are peculiarities to a model that surely does not accurately reflect any natural population. So the advance made here is quite modest. A paper with this scope seems much better suited to a journal specialized in genetic models (e.g. TPB, JMB, etc.).

A final weakness of the paper is the scholarship. A couple of minutes on Google Scholar shows that quite a few genetic models for the evolution of assortative mating are not cited here. This literature is not enormous, and any theoretical paper on the topic really must do a thorough review to put its results into context.

Response to Referee #1 (Dr. Michael Kopp)

General response:

We thank Dr. Michael Kopp for providing plenty of constructive criticism, which we appreciate a lot. Our revised work takes his comments fully into account.

Point-by-point responses:

Comment 1: The writing needs serious improvement. I have extracted the main text from the pdf and attach a .doc-file with (lots of) suggestions for revision (as well as comments; I didn't have time to do the same for the Appendices and supplementary Figures). I apologize to the authors if this comes across as condescending, which is not my intention. Of course, these suggestions are not obligatory.

Response 1: We thank Dr. Michael Kopp for all those suggestions – the work involved must have been significant. We have revised the manuscript with all this in mind. Below, in the section “minor points”, we specifically answer to minor comments raised in the .doc-file.

Comment 2a: A potential limitation of the results is that 'preference cycling' only occurs if (male?) choosiness is nearly perfect. According to Figure S8, preference cycling vanishes if $\epsilon = 0.03$, that is, if choosy males have a three percent probability of courting a non-matching female. In this case, female choosiness remains under selection and does not degrade due to drift. One might wonder if $\epsilon = 0.01$ (as in most simulations) is realistic in nature, unless subpopulations are already strongly diverged. At the very least, this requires some discussion.

Response 2a: This is an excellent point. By reading the previous version of the manuscript, one might conclude that preference cycling only occurs if '*choosiness is nearly perfect*'. This would considerably reduce the breadth of interest in our findings. In the revised version of the manuscript, we highlight that preference cycling is not limited to this extreme scenario and that our predictions may apply to a large variety of ecological contexts.

Specifically, preference cycling only occurs if choosiness leads to a '*nearly perfect reproductive isolation*' (page 14, lines 341-345). Speciation is often associated with the emergence of a large variety of isolating barriers, including premating isolation. Speciation might only occur when the isolation caused by all isolating barriers is nearly complete. In additional simulations, we assume that other isolating barriers have evolved such that, without choosiness expressed during mating, the two ecotypes are partially reproductively isolated (figure S14). Under this scenario under which speciation appears very likely, we show that preference cycling increases hybridization rate.

Comment 2b: Regarding the last point, I wonder if preference cycling might be more prevalent if female choosiness is costly (the authors briefly discuss costs for females in Appendix A, and conclude that the results are not qualitatively altered if females can reallocate at least 85% of rejected mating opportunities to preferred males). In this case, female choosiness might become neutral or deleterious already at lower values of male choosiness. Indeed, if choosiness is costly for both sexes, an almost game-theoretical situation arises, where both sexes benefit from assortative mating in terms of offspring fitness, but would prefer it to be taken care of by the other sex (not expressing choosiness

would then be akin to cheating). Whether or not such a scenario arises might depend on the exact form of costs.

Response 2b: In **Appendix A**, we now provide a thorough analysis of the implication of costs of female choosiness in preference cycling. Specifically, we show that such game-theoretical situation may arise. Female choosiness may be more deleterious when males are choosy. However, we also show that positive frequency-dependent selection caused by female choosiness is necessary for further female choosiness to evolve when males are nonchoosy (as shown by Otto et al., 2008). Consequently, starting from a random mating state, female choosiness does not evolve; preference cycling is not more prevalent if female choosiness is costly.

Even if female choosiness is costly, preference cycling is induced by genetic drift and completed by selection.

Comment 3: One issue I do not understand is negative indirect sexual selection on female choosiness as shown in Figure 2 ("sex-"), which is responsible for the male-choice and partial male-choice regimes in the deterministic model. Indirect negative sexual selection on female choosiness means that choosy females will have choosy sons (due to LD between M and F), whose mating success will be low (because of lost mating opportunities and high competition). This should happen if there are too many choosy males relative to the number of choosy females. However, there is one thing I don't understand at all: How is it possible that there are "sex-" regions for females that overlap with "sex+" regions for males (Fig. 2c-e)? Since sexual selection on females can only be indirect (as all females have by definition equal mating success), how can it be negative on females when it is positive on males? This needs to be clarified. Also, explain in a bit more detail what happens in Figures 2d and 2e (see my comments in the attached file).

Response 3: Indirect sexual selection acting on female choosiness does not necessarily reflect direct sexual selection acting on male choosiness, as indirect selection is determined by three-locus linkage disequilibrium between loci coding for the ecological trait, female choosiness and male choosiness. In particular, choosy males carrying alleles that code for female choosiness are mostly homozygous at the ecological locus (figure S2). While choosy heterozygous males benefit from reduced male-male competition because heterozygous females are the least popular ones (contributing to "sex+" regarding male choosiness), choosy homozygous males carrying alleles coding for female choosiness suffer from male-male competition (strongly contributing to "sex-" regarding female choosiness). Consequently, sexual selection can favour male choosiness (sex+) while simultaneously inhibiting female choosiness (sex-). This property is responsible for the male-choice and partial male-choice regimes in the deterministic model. This is now clarified (**page 9, lines 228-235**).

Comment 4: Model with continuous choosiness (Appendix B): Are the low but positive fixation probabilities in Fig. B1 due to selection or drift? I suppose you need to take into account chance fixation of neutral or deleterious alleles in order to model genetic drift? Most importantly, what is happening with male choosiness for $\alpha = 0$? Why does it fluctuate quasi-periodically? Shouldn't it be selected against?

Response 4: In those simulations, genetic drift can oppose selection and neutral or deleterious alleles can get fixed by chance (e.g., fixation alleles F_{i-1} when males are choosy in Fig. B1) (**caption of figure B1**). In Fig. B2, partial male choosiness evolves by chance because male choosiness is little deleterious when females are strongly choosy (cf. Fig. 4a); selection is strong enough to inhibit the evolution of strong male choosiness

(caption of figure B2). In additional simulations, we show that there is no such fluctuation of male choosiness for $\alpha=0$ if there is little genetic drift ($K=1,000$, figure B2)

Comment 5: You say that you always use values of s (strength of disruptive selection) that assure that homozygotes are more frequent than heterozygotes, such that heterozygotes are never favored by sexual selection. However, this cannot be true in, e.g., Figure 2, where you explore all values of s between 0 and 1. Does sexual selection for heterozygotes play a role in these cases?

Response 5: We made an error. In our model, sexual selection induced by female preferences does not hamper ecological divergence. This is not caused by the high frequency of homozygotes at the ecological locus; indeed, heterozygotes are often in higher frequency than homozygotes (especially if s is low). However, if heterozygotes Aa are initially in high frequency, positive frequency-dependent sexual selection on the A locus increases the mating success of those heterozygotes but does not increase their frequencies (crosses $Aa \times Aa$, $Aa \times AA$, $Aa \times aa$ all lead to 50% Aa offspring). Therefore, sexual selection does not increase the frequency of heterozygotes at the A locus and does not inhibit ecological divergence; see page 7, lines 174-177 for how we now discuss this topic.

Comment 6: Can you say a bit more about possible effect of your choice of recessive choosiness alleles? For example, this should affect the fixation probabilities in Appendix B.

Response 6: We conducted additional simulations to test the effect of dominance hierarchy on the evolution of choosiness. We get qualitatively similar deterministic equilibria when alleles coding for choosiness are recessive (figure 2) or dominant (figure S5). Quantitative differences rely on the initial conditions. We implement initially 1% of choosy individuals in all deterministic simulations (page 7, lines 168-169). Therefore, initial frequencies of alleles F and M differ according to the dominance hierarchy implemented.

In stochastic simulations, preference cycling is less likely if alleles coding for choosiness are dominant (page 14, lines 336-338; figure S11). Once advantageous choosiness alleles have reached a high frequency, recessive nonchoosiness alleles are necessarily rare and thus mostly present in heterozygotes, such that complete fixation of the dominant choosiness allele is difficult. Therefore, in stochastic simulations, *reproductive isolation is rarely nearly perfect* if alleles coding for choosiness are dominant. This is not a condition favouring preference cycling (see Response 1).

Comment 7: In the introduction and/or discussion, say a bit more about the results of previous models of speciation with male or mutual mate choice (e.g., ref. 38).

Response 7: In the Introduction, we now specify a key result concerning the implication of male mate choice evolution during speciation. Male mate choice is less likely to evolve and to contribute to reproductive isolation than female mate choice (male-male competition generates sexual selection against male choosiness) (page 3, lines 49-50). In the Discussion, we highlight that this prediction does not hold once preference coevolution can occur; female choosiness favours the evolution of male choosiness (page 16, lines 377-380). Regarding Ref. 38 in particular: it is an excellent review of male and mutual mate choice, but not in the specific context of speciation; we therefore find it appropriate to cite it but not spend much more space discussing its findings.

Comment 8: The effects of preference cycling on speciation will depend on its timescale (e.g., it is most severe when the population stays in the P regime for a long time). Can you say a bit more about this? How long does one cycle take, i.e., what is the mean time between episodes of hybridization? Is it so long that other isolating barriers might conceivably already have evolved?

Response 8: Barriers to gene flow are likely to accumulate with time and we agree the period of preference cycling may affect the accumulation of other barriers. We provide the probability of initiating a cycle, which provides a partial answer to this. However, the rate at which other barriers, involving other processes than mate choice, might accumulate in between cycles of preference is unknown, and of course beyond the scope of this model because it requires specific assumptions about the nature of isolating barriers. However, this is an interesting point which we now discuss as a perspective for future work in our Discussion (page 18, lines 442-444).

Comment 9: What do you think happens if loci are more tightly linked ($r < 0.5$)?

Response 9: Unlike preference/trait rules, under which assortative mating depends on the maintenance of genetic polymorphism at a minimum of two distinct loci (ecological and preference loci), we model here matching rules under which the same choosiness allele gets fixed in both ecotypes. Consequently, physical linkage among each choosiness locus and the ecological locus is unlikely to affect our results. For simplicity, we also did not consider physical linkage among choosiness loci (this restrains the number of genotypes to follow, see response to Referee 2). We do not have the intuition about how physical linkage between alleles coding for choosiness may alter our results. Given the extent of analyses it would require, we did not run further simulations to explore this aspect. However, we suggest this as a theoretical perspective to explore (page 18, line 440).

Minor points:

=> We agree that viability selection is frequency-dependent; relative frequencies of homozygotes affect viability selection (relatively to parameter s'). This component of viability selection allows the maintenance of polymorphism at the ecological locus but it does not impact indirect selection acting on choosiness. Given that our paper focuses on the evolution of choosiness, we chose not to specify this feature of viability selection in the title of the subsection describing viability selection (Methods) (as suggested) because it might confuse the reader. Note, however, that this feature of the model is clearly described in the Methods.

=> We did not switch the subscripts of $P^{\text{male}}_{\{m,f\}}$ and $P^{\text{female}}_{\{f,m\}}$, the first subscript (f or m) representing the sex of the individual expressing a preference (or not). In the first case it is the male (subscript m) whereas, in the second case, it is the female (subscript f).

=> Colourations red, blue, purple and black are consistently used in the manuscript to describe equivalent choosiness regimes. Therefore, we did not change the colouration of fig 2 and 5 in the revised version of the manuscript, as suggested.

=> To conduct stochastic simulations, we sample offspring using the theoretical distribution of genotype frequencies. Given that we do not sample individuals from a larger population, replacement is not an option.

Response to Referee #2

General response:

Referee 2 appears to argue that we claim that hybridization rate can increase due to mutual mate choice (a direct causality). This is not how we intend our results to be read; our argument is more subtle. The direct causality is the usual one: stable mutual mate choice reduces hybridization; but we add the important distinction that mutual mate choice is not as stable as one might think. Thus, when we allow for coevolution of male and female choice, mutual mate choice is easily destabilized, and this destabilization then leads to increased hybridization. Referee #1 expressed this very succinctly, "This pattern of preference cycling (triggered by drift, but completed by selection), which induces periodic episodes of hybridization, is the key result of this paper".

As a whole, using coevolutionary simulations, we describe a new phenomenon that is not found when one sex alone is allowed to evolve mate choice (the latter being amply covered by the rich literature on the subject); and we explore the consequences of this on speciation.

We find Referee 2's request for additional results with choosiness expressed in one sex alone to be very useful to highlight how precisely mutual mate choice can destabilize itself, also for the purposes of clarifying our argumentation above. We show that premating isolation is unstable only when both sexes can contribute to it, because mutual mate choice is inherently unstable in face of drift, even when selection tends to favour it.

Point-by-point responses:

Comment 1: In the stochastic model, the frequency of heterozygotes (hybridization rate) increases under some conditions, which, the authors argue, is caused by mutual mate choice. However, if the preference loci are absent (no mate choice), the hybridization rate should be almost 0.5 before selection, which is clearly higher than that under mutual mate choice. Therefore, some presentations (e.g., title: Coevolution of male and female mate choice can destabilize reproductive isolation) are misleading because assortative mate choice necessarily decreases the hybridization rate. I guess that the authors want to argue that mutual mate choice is less effective for reproductive isolation than mate choice by single sex. If so, they should compare the hybridization rate under mutual mate choice with those under single mate choices. It is easy to make the model simulation as female choosiness alone and male choosiness alone (keep the frequency of M or F as zero). Without the comparisons, their argument is never confirmed.

Response 1:

As noted by Referee 2 and as was shown in the previous version of the manuscript (**page 11, lines 266-268 in the revised manuscript; figure S4**), stable mutual mate choice leads to a very low hybridization rate in deterministic simulations. However, in our paper, we describe a counterintuitive coevolutionary dynamics triggered by drift *under selection favouring mutual mate choice*. In stochastic simulations, the regime of choosiness can evolve away from mutual mate choice, leading to "preference cycling" (**page 11, line 281**). So increased hybridization is due to the time spent *away* from mutual mate choice, because of preference cycling. As a consequence, reproductive isolation can be considered easily destabilized *because of preference coevolution*. Comparing deterministic equilibria (when preference cycling never occurs) and stochastic simulations

(when preference cycling occurs) confirms that this dynamics of coevolution triggered by drift increases hybridization rate. In the Discussion, we now emphasize that mutual mate choice *per se* leads to strong reproductive isolation, but that it is particularly unstable in finite population (**page 16, lines 379-381**).

Following the Referee's comment, we now compare time series of choosiness evolution and resulting hybridization rates when one sex vs. both sexes can evolve choosiness (**figure S7**). This figures makes it clear that preference coevolution destabilizes reproductive isolation periodically, increasing hybridization rate; it is not mutual mate choice *per se* that increases hybridization rate, but its frequent breakdown. Additionally, we provide the main sensitivity analysis of our paper, this time when one sex can evolve choosiness (**figure S8, analogous to figure 5d**), showing that preference coevolution occurring when choosiness can evolve in both sexes leads to passage in a regime of random mating and increases hybridization rate. We hope these supplementary figures make our case clearer.

Response to Referee #3

General response:

We find Referee 3's language rather strong, and we here explain why our views differ from those of the Referee. We hope to provide a convincing argument that this manuscript, especially in its revised form, is a significant advancement for the field as a whole.

Point-by-point responses:

Comment 1: The foundational problem involves the genetic bookkeeping. The model tracks the frequencies of the three genotypes at each of three loci, for a total of $3^3 = 27$ dynamic variables (l. 117-120). But in fact this genetic system has 36 dynamic variables. Take the simpler case of just two loci, A and B. The frequencies we need to follow are those of all the diploid genotypes, keeping track of which alleles were inherited together from the same gamete:

ab/ab, ab/aB, ab/Ab, ab/AB,
aB/aB, aB/Ab, aB/aB,
Ab/Ab, Ab/AB,
AB/AB

Thus there are 10 frequencies, rather than the $3^2 = 9$ frequencies that the ms would claim. In general, with n diallelic loci, there are 2^n types of gametes. The total number of diploid genotypes is

$2^n + (2^n - 1) + \dots = 2^{n-1} (2^n + 1)$,
not 3^n as claimed in the ms.

Thus it appears that the recursion equations that are the basis for this study are wrong. It is possible that doing the model correctly will lead to similar conclusions, but that is unknown. I believe that the only case in which the ms' equations predict the dynamic behaviour correctly is when there is no linkage disequilibrium. The nonrandom mating in this model would seem to generate linkage disequilibrium, and so I suspect the dynamics are wrong.

Response 1:

According to the Referee, our genetic system should have 36 dynamic variables and not 27. This would be a very gross mistake indeed, and the Referee used accordingly strong language: "fundamental flaw in its mathematical foundation...". However, the Referee's argument is only true when loci are in physical linkage on the chromosome, preventing them segregating freely in the gametes, and necessitating separate variables for each association of alleles. The Referee indeed explains the point with an example with physical linkage. But this argument does not hold when loci are unlinked, which was our assumption (we explicitly state 'Mendelian inheritance of all loci with no physical linkage; **page 4, line 88**'). In this case alleles at each locus segregate during meiosis, which reduces the number of variables needed for the bookkeeping.

The details of the argument are as follows. In gametes, recombination can break down associations of alleles at different loci with a recombination rate = 0.5. Alleles assort independently of one another in gametes following Mendel's second law and, therefore, both alleles of a given locus have equal probability of being associated with the alleles of unlinked loci during meiosis. This is the key process that allows the reduction of the number of dynamic variables of our genetic system from 36 to 27. Put differently, we can

say that among the 36 dynamic variables, 9 are redundant because of free recombination during meiosis.

To make it absolutely clear that we are right, consider the simple case of two loci A and B taken by the Referee:

Individuals of genotype **ab/AB** generate gametes **ab** and **AB** (no recombination) and gametes **aB** and **Ab** (recombinant gametes)

Individuals of genotype **aB/Ab** generates gametes **aB** and **Ab** (no recombination) and gametes **ab** and **AB** (recombinant gametes)

If recombination rate is 50% (unlinked loci) those two parental genotypes produce rigorously the same distribution of gametes and we can model them as a single dynamic variable: genotypes **Aa-Bb**. Under free recombination and with two loci, we can therefore reduce the number of dynamic variables from 10 to 9. With three loci, the reduction is from 36 to 27.

Note, however, that genetic disequilibrium can arise without physical linkage (page 8, lines 180-188). In particular, the coevolutionary dynamics that we describe relies on nonrandom mating generating linkage disequilibrium. Selection causes linkage disequilibrium to arise; some combinations of alleles (e.g., genotypes FF-MM-AA and FF-MM-aa) are more frequent than expected at random. Such linkage disequilibrium among unlinked choosiness and ecological loci explains the emergence of premating isolation in numerous seminal theoretical models of assortative mating (most of them being cited in our manuscript).

Our conclusions and discussion of linkage disequilibrium may have contributed to the Referee's assumption that there must be physical linkage in the model. However, linkage disequilibrium can be generated by selection and by migration in the absence of physical linkage, and our discussion of linkage refers to this process. To avoid any future reader being misled this way, we now explain more clearly why we can reduce the number of variables from 36 to 27 due to the modelling of unlinked loci (page 5, lines 102-106).

The reduction of the number of variables to be kept track of is clearly an advantage of our approach, but it could also be seen to be a limitation, and we now acknowledge it as a limit of the model (page 18, line 440). Here, one option could have been to perform a more complete analysis of genetics where physical linkage is also taken into account. However, as we were also asked to perform various additional simulations due to the constructive criticism of Referees #1 and #2, we have not opted for yet another approach in the current context, but simply highlight explicitly that this assumption has been made.

Comment 2: A second consideration makes me feel that this paper would not be a good fit for Nature Communications even if the math was right. The model is not biologically general: the mating rules, number of loci, dominance relations, and lack of pleiotropy in the behaviour of males and females are just some of the limitations. Unfortunately, the paper includes only simulations -- general analytic results are entirely lacking. So it is not clear which results are likely to be biologically robust, and which are peculiarities to a model that surely does not accurately reflect any natural population. So the advance made here is quite modest. A paper with this scope seems much better suited to a journal specialized in genetic models (e.g. TPB, JMB, etc.).

Response 2:

All models have limitations — the quest of a modeller is to seek a balance between tractability and the number of interesting phenomena that can be included. As we

explained above, we could have dealt with comment 1 by including ever more detail (considering physical linkage too), but this clearly contradicts with the request to consider analytical resolution too. It is well-known that solving analytically a system with three independent loci is very problematic (cf. Servedio & Lande 2006, which we cite in our manuscript; page 4, line 89). Tracking the linkage disequilibrium between three loci is fastidious; most population-genetic models only track the genotypes of two loci. That is why solving our genetic system is, unfortunately, not within reach (just like Servedio & Lande 2006 showed for their model).

More generally, a criticism about lack of realism due to a limited set of parameters can be made on nearly all models that can be found in the literature, and which are all simplifications of the real world. Yet, simple mathematical models and especially population genetics models with single loci coding for complex traits (like mate choice) have been very helpful to decipher complex evolutionary phenomena (cf. Felsenstein 1981 on speciation or Kirkpatrick 1982 on sexual selection, cited in our manuscript). It seems therefore unfair to discredit our approach, especially as it deals with a process where temporal dynamics interacts with drift in a fundamentally stochastic manner, revealing novel kinds of dynamics.

We agree, however, that it is absolutely crucial to discuss the limits of our models, and to what extent our conclusions hold with more complex features (as also highlighted by Referee #1) and we have improved this aspect in a revised version of the manuscript (e.g. Appendix A and B, and supplementary figures). In particular, implementing more loci (e.g., changing the mating rule into a preference/trait rule and increasing the number of loci) would be very complicated for the reasons outlined above. However, we now suggest such explorations as theoretical perspectives (page 18, lines 439-441).

Finally, following the comment made by Referees #3 and #1, we now analyse the effect of dominance hierarchy in the revised version of the manuscript (see Response 6 made to Referee #1). On the other hand, we did not implement pleiotropy in the behaviour of males and females for the simple reason that we need to implement independent loci to study the coevolution occurring among traits. Pleiotropy also occurs in nature, but this is not the situation we wanted to model.

Please note that, at present, the extent of our sensitivity analyses is considerable:

- carrying capacity
- strength of viability selection
- strength of choosiness
- implication of other isolating barriers
- continuous choosiness
- cost of male choosiness (lost courtship opportunities)
- cost of female choosiness (lost mating opportunities)
- implementation of neutral loci
- hierarchy dominance

Of course, one can always relax more and more assumptions; one has to seek a balance between this and choose judiciously what is best left for future study. We hope that the present analysis captures well the determinants of preference coevolution and the robustness of the key finding.

Comment 3: A final weakness of the paper is the scholarship. A couple of minutes on Google Scholar shows that quite a few genetic models for the evolution of assortative mating are not cited here. This literature is not enormous, and any theoretical paper on the topic really must do a thorough review to put its results into context.

Response 3:

We are thoroughly aware of, and appreciate, the vastness of the literature on assortative mating. We believe that the task of authors of a paper is not to prove they have found them all (we in any case had already reached the Nature Comms limit of 70 citations), but to do a proper job, helpful for the reader, when choosing the most relevant ones that help to put our study into context. We believe we have chosen the most relevant literature to cite by focusing on studies that ask specifically how the mating pattern impacts prospects of speciation. Without precise suggestions of which paper would be relevant enough to kick one of those out, it is difficult for us to improve the list.

Reviewers' Comments:

Reviewer #1:

Remarks to the Author:

I have reviewed a previous version of this manuscript, which was originally rejected before finally receiving an invitation for revision. I am still very positive about this paper, and think that it would make an important contribution to Nature Communications. I have to say that, given the amount of effort I put into my first review, I was somewhat taken aback by the way the other two reviewers waved off the paper in a just a few paragraphs.

The authors have produced a very thorough revision and responded to all questions and criticisms. While I was quite critical of the writing of the first version, the second version is considerably improved. My long list of editorial comments below might seem to contradict this statements, but the suggested improvement are much more minor than the first time around. I still have a few questions to specific details, listed below, which I am sure the authors will be able to address.

I therefore recommend acceptance of the manuscript after some more minor revisions.

Signed,

Michael Kopp.

Specific comments:

- 72: one-allele mechanisms is unclear here, and might best be simply not mentioned (what is a one-allele mechanisms is the evolution of increased choosiness [in either sex] in both AA and aa subpopulations)
- 87 and elsewhere (e.g. 309): Strictly speaking, polygyny does not mean (nor necessarily imply) that all females have equal mating success, but only that one male can fertilize several females.
- 100: Is homotypic preference the same as phenotype matching? If yes, it might be good not to mix these two terms; the first parenthetical in this line might also read "magic trait"
- 120: Without negative frequency-dependence, swamping will occur even under random mating [this sentence and the next one might also simply be deleted]
- 154: This is true only for weak costs of choosiness!
- 168, one percent of choosy individuals are introduced: does this also hold for the stochastic model, or do you wait for mutations in this case?
- 169: I assume also LD between F and M loci, and Hardy-Weinberg at both?
- 171: is μ the mutation probability per allele or per (diploid) locus? (a common source of confusion!)
- 172, "by averaging over these runs": and over time within runs?
- 174-177: I don't quite understand this argument (see also Fig. S3). In a system with two alleles, heterozygotes always have 50% of heterozygous offspring, no matter what. If they have less mating success, shouldn't they still produce less (homozygous or heterozygous) offspring? Note that ref. [22] uses essentially the same model of female choice, and in that paper, sexual selection favoring heterozygotes can prevent the evolution of choosiness (and hence what you call ecological divergence; indeed, the situation you describe seems to be equivalent to the R/C regime in [22]). I wonder if what happens in your model is the following: You introduce a non-negligible proportion (1%) of choosy females; while almost 50% of the population are still heterozygous at the ecological locus, the choosiness allele rapidly develops strong LD with homozygous genotypes at that locus; in consequence most of the `_choosy_` females actually prefer homozygous males, and this creates sexual

selection in favor of homozygotes, and hence does not impede the spread of the choosiness allele.

- 228: Maybe: "In the simplest case, indirect sexual selection on female choosiness may be seen as reflection of direct sexual selection on male choosiness, which is transmitted via LD between the M and F loci. However, this is not necessarily so, as evidenced by parameter regions of female sex. - that overlap with male sex. + in Fig. 2d-f. Indeed, net indirect selection results from LD between all three loci. In particular, choosy males that also carry alleles for female choosiness (genotypes FF-MM) are particularly likely to be homozygous at the ecological locus, whereas choosy males not carrying such alleles (genotypes ff-MM) are somewhat more likely to be heterozygotes Aa [Is this true? I find it hard to judge from Figure S2; also, this Figure does not show results for heterozygous choosiness genotypes Ff or Mm]). While choosy heterozygous males benefit from reduced male-male competition (contributing to sex. + on male choosiness), choosy homozygous males (which are strongly linked to female choosiness) face intensified male-male competition (strongly contributing to sex. - on female choosiness). As a whole, sexual can therefore favor male choosiness and inhibit female choosiness.
- 348: What exactly is the probability to reach regime P? Do you start repeated simulations in regimes F and FM, and count how many times the simulation enters P? But won't it always reach this regime eventually, if you wait long enough?
- Fig. S7+8: point out somewhere in the main text that male choosiness alone can evolve only if alpha is very high (about 0.9)
- 336 and Fig. S11: I don't get the argument for why dominant choosiness alleles reduce the likelihood of preference cycling. You say that fixation of the dominant allele is difficult, because the recessive allele, once it is rare, is no longer exposed to selection. But at the phenotypic level, this means that the proportion of choosy males becomes very high (my guess is as high as in the recessive model), so selection on female choosiness should vanish, drift should take over etc.
- 341 and Fig. S11: I think I get your argument here: Even if choosiness is not perfect, strong choosiness in combination with other barriers might make reproductive isolation perfect enough to trigger preference cycling. Two comments though: First, I don't understand at all what $\min(P_{\{m,f\}}^{\text{male}}) = \min(P_{\{f,m\}}^{\text{female}})$ means in Fig. S11. Logically, the presence of other barriers should mean that reproductive isolation can never drop below a certain value, even if choosiness evolves to zero, but I don't see how this idea is conveyed by the above condition. Second, the fact that RI cannot drop below a certain value means that the "amplitude" of preference cycles is limited, and hybridisation should stay lower. This effect should be briefly discussed.
- 391: Incompatibilities can arise in the presence of gene flow if the alleles fixing in the two populations are favored by selection (see Bank, Bürger, Hermisson 2012, Genetics)
- Appendix A: I don't quite understand the second-to-last paragraph, in particular the part about cheating and positive frequency-dependence (see also my editorial comments below).
- Fig. S7 shows that male choosiness alone can evolve only for large values of alpha. This should be pointed out in the main text (when describing Fig. 1).

Editorial comments:

- 5: (assuming a phenotype-matching rule)
- 7: with reciprocal preferences
- 9: "rate" can be omitted
- 10: rather fragile
- 12: by different isolating barriers
- 13: sexual isolation instead of premating isolation?
- 24: such indirect selection can favor
- 24: stronger homotypic preferences
- 25: often induces (or: many forms of assortative mating induce)
- 30: inhibit the initial evolution
- 31: it is not entirely clear that you have been talking about magic traits here; maybe specify "When

mate choice is based on phenotypic traits under divergent or disruptive ecological selection (so-called magic traits)" already in line 22; in this case, omit "such" here

- 33: when mate choice can be expressed
- 35: in both females and males
- 39: for gaining a mate
- 49: at least as long as it does not generate competitive or opportunity costs
- 71: which are
- 75: caused by either
- 77: How about: Each generation first experiences/undergoes disruptive viability selection?
- 78: suffer increased mortality
- 84: Females likewise
- 87: The expected genotype frequencies
- 87: maybe: depend on the probabilities of mating between different genotypes
- 88: being formed assuming independent Mendelian inheritance at all loci
- 89: of mutual mate choice
- 94: just as
- 95: in these stochastic simulations, we allow for mutation of alleles in offspring
- 104: This assumption
- 104: the number of dynamic variables needed to describe our genetic system
- 113: favour
- 114: To ensure the maintenance of polymorphism, a parameter $s' > 0$
- 115: viability costs described by a parameter s . Following ...
- 137: part of the saved courtship effort
- 145: locus "F" in italic
- 150: analogous
- 151: that are less preferred by males (or that prefer rare males) ... (no cost of choosiness and no sexual selection from male choice)
- 152: with and without preference is identical
- 154: $f = FF$ (or $f \in \{FF\}$)
- 156: are calculated
- 161: yielding the expected
- 163: "K" in italic
- 166: strong choosiness in genotypes ...
- 169: in such a way that
- 172: For each parameter combination
- 174: by female choosiness
- 177: add a reference to Fig. S3 here
- 182: omit "non-random"
- 186: to avoid repeating "this linkage disequilibrium" you might start the sentence simply with "In consequence, ..."
- 191: In addition, female choosiness can induce ... on this locus, such that homozygous males have the highest mating success.
- 203: which further lowers
- 205: is favored only if
- 212: also indirectly affect
- 213: Should the dashed arrows in Fig. 1c point in both directions (because of the vice versa)?
- 219: I don't see any arrows with low opacity in fig. 1c. Is this sentence still relevant?
- 224: enforce the evolution of male choosiness in situations where it would otherwise not evolve
- 225: maybe add "direction of (indirect) sexual selection on female choosiness"
- 223: the notation $sex. +$ and $sex. -$ looks really weird in the main text; can you change it to something like $S+$ and $S-$ or $sex+$ and $sex-$ and maybe italicize it or put it in quotes?

- 226: in particular, male choosiness clearly ...
- 230: loci
- 243: after mating and reproduction
- 251: are at low risk of producing unfit hybrids
- 255: will be assessed below
- 261: AM is based only on
- 268: simply: Changing the dominance hierarchy ...
- 272: We next ran
- 274: scenarios with strong disruptive selection ($s = 0.2$), for which the deterministic outcome is mutual mate choice.
- 277: are at most partly [or partially] choosy
- 282: For $\alpha > 0.01$ (and strong selection)
- 289: measured after the equilibrium ... has been reached for the first time
- 291: In stochastic simulations, in contrast, assortative mating is often based on male choosiness only
- 294: based on male choosiness
- 297: may then
- 298: the end of this sentence comes across as somewhat repetitive, since the fact that AM is based on male choosiness is what you set out to explain just 5 lines prior; but spontaneously, I don't have a good alternative suggestion.
- 299: The partial choosiness regime (P) is rare for both low and high values of α . When $\alpha < 0.01$, male choosiness does not evolve and female choosiness is under sufficiently strong selection to remain at high frequency (regime F). When $\alpha > 0.9$, choosy males can reallocate most of their courtship effort, and male choosiness is maintained by indirect natural selection even if female choosiness (and the associated sexual selection) is absent.
- 302: simultaneously partly choosy
- 302: The reason is that, for intermediate α , male choosiness is favored only when a large proportion of females are choosy. When female choosiness decreases due to drift, male choosiness becomes selected against, and the population can enter the partial choosiness regime. Although selection (first for female and then for male choosiness) will ultimately cause a return to mutual mate choice, this process takes time, and a snapshot of the population at a given point in time has a significant chance of observing the P regime.
- 306: Hereafter, we refer to this coevolutionary dynamics as preference cycling, since female and male choosiness go through ...
- 310: or if choosiness is allowed to vary as a continuous trait
- 316: over one entire generation (net effect of natural and sexual selection combined) in the deterministic model, and are therefore simpler than the representation in Fig. 2b-f.
- 317: the length of arrow vectors is drawn on a logarithmic scale
- 321: whose direction
- 322: Since we assume that neither sex can ever
- 323: is lowest
- 324: and becomes somewhat higher
- 330: e.g., fluctuations of F_{st} measured at neutral loci between genotypes AA and aa
- 330: even though mutual mate choice, whenever it occurs, achieves the strongest degree of assortative mating, it is also particularly prone to periodic break-downs, which as a whole, hamper the maintenance of pre-mating isolation. Importantly, these periodic breakdowns of assortative mating / reproductive isolation do not occur if choosiness is possible [not favored!] in one sex only.
- 338: less perfect
- 341: if the evolution of choosiness leads to nearly perfect RI between ecotypes. This is the case if choosiness per se is nearly perfect, or if reproductive isolation is strengthened by additional barriers to gene flow. Under these conditions, which intuitively, seem most conducive for speciation, we show that

coevolution ... has the potential to ...

- 351: maybe "in previous models of speciation with gene flow" to prevent the repetition of "reproductive isolation" at the end of the next sentence
- 363 and elsewhere: You might want to think of when to use "preference" and when "choosiness". At the moment, you seem to use these two terms interchangeably and more or less at random.
- 366: have
- 368: The sentence "Consider ..." is unnecessary, as the following argument does not rely on the absence of female choosiness
- 369: because even non-choosy females can avoid
- 370: The sentence "Now consider..." can be omitted
- 371: While female choosiness reduces indirect selection for male choosiness in a similar manner, it also strongly favours male choosiness through direct sexual selection. This is because male choosiness is a poor strategy ... If female are choosy, however, choosy males gain a high mating success b disproportionately ...
- 377: "Under disruptive viability selection" is really out comes context here. I would write: Therefore, male choosiness evolves easier in our model than in models investigating the evolution of choosiness in each sex separately. Indeed, our results show that mutual mate choice should often be favored by selection, and can induce very strong reproductive isolation in infinite populations.
- 381: Counterintuitively, however, in finite populations, mutual mate choice is particularly unstable. In the presence of even weak genetic drift...
- 386: male choosiness becomes disfavored by selection, temporarily leading ...
- 389: comma after speciation
- 390: a key prerequisite
- 390: allows the establishment
- 391: selection for
- 398: To track the speciation continuum, empirical research...
- 399: snapshots in time
- 401: mean
- 403: a species'
- 412: which also affect
- 415: with even little reallocation; without reallocation, in contrast, male choice should be deleterious. Importantly, if reallocation is partial,
- 420: each sex relies
- 433: In this context
- 433: preference cycling could likewise lead to temporal increases ...
- 438: against hybrids
- 439: a more detailed look
- 443: other barriers
- Appendix A: In the main text, we assume that all females are guaranteed to mate. ... This corresponds to the polygyny scenario ... For instance, this corresponds ... [no new paragraph after "very costly"] ... If costs of female choosiness are high (many lost mating opportunities, $\beta < 0.85$), female choosiness does not evolve even under strong disruptive viability selection ($s = 0.2$) and in the absence of male choosiness. Consequently, male choosiness does not evolve either (because in the absence of female choosiness, it is disfavored by direct sexual selection) and the population remains in a state of random mating. If female choosiness is weak ($\beta > 0.85$, ... qualitatively similar to those without costs of choosiness. In particular, even if female choosiness is costly, preference cycling is still induced by genetic drift and completed by selection.
- Fig. A1: In particular, preference cycling is not more prevalent if female choosiness is costly; the initial decline of female choosiness still needs to be induced by genetic drift, not by selection against costly choice. [But I'm not sure where I can see this in the Figure.]
- Appendix B: ... choosiness is therefore a binary trait ... we here place our model in the adaptive-

dynamics framework ... mutate to the closest neighboring allele (stepwise-mutation model) ... by mutants carrying ... probability of invasion for each allele ... Using these invasion probabilities ... these evolutionary dynamics ..

- Fig. B2: is only weakly deleterious
- Fig. S2: I don't think one can write "linkages" instead of "linkage disequilibria"
- Fig. S2: between two diploid genotypes ... female and male choosiness genotypes ... vanish ... under this condition
- Fig. S3: this would not be case ... completely reallocate
- Fig. S5: similar to the one obtained with recessive choosiness alleles ... and so does the deterministic equilibrium reached (in cases with more than one stable equilibrium)
- Fig. S6: The description of the modeling approach at the beginning of the caption is difficult to understand. Can't you just say that you added 20 unlinked neutral loci to your simulation model?
- Fig. S6: We compute the F_{st} statistic as a measure of neutral genetic differentiation between ecotypes ... male and female choosiness favour ... genetic differentiation is stable [I suppose heterozygotes are ignored for this calculation?]
- Fig. S7 + S8: IF choosiness evolves in both sexes, the resulting coevolutionary dynamics destabilizes reproductive isolation.
- Fig. S9: rarely evolves ... (small blue area), and preference cycling never occurs
- Fig. S10: ... preference cycling does not occur if drift is weak, as female choosiness rarely (almost never) drops to values low enough to induce selection against male choosiness / to change the direction of selection on male choosiness.
- Fig. S11: the insert "(here, viability selection is weak; $s = 0.1$)" is repetitive; replace "For this dominance hierarchy" by "With dominant choosiness alleles"; also see my comments above
- Fig. S14: Maybe prefer "Effect of additional isolating barriers", and see my comments above

Reviewer #2:

Remarks to the Author:

I previously commented that the argument did not correspond to the analysis. In the revised manuscript, the argument becomes clearer and I realized that I somewhat misunderstood the argument. But therefore, I also realized that I somewhat overestimated the value of the paper, because I previously thought that the paper showed that mutual mate choice is less effective for reproductive isolation than mate choice by single sex. The authors merely argue that reproductive isolation is sometimes destabilized by mutual mate choice. Since reproductive isolation is easily destabilized when preference loci disappear, which may occur by genetic drift, the result is not impressive enough. The authors also emphasize that preference cycling occurs when considering mutual mate choice, but this result is theoretically not surprising because it occurs in simpler sexual selection models, and biologically not significant enough because no empirical data show this phenomenon (if existed, please raise a reference). Nevertheless, I recommend this paper for publication after a revision, because this paper seems to show that mutual mate choice is less effective for reproductive isolation than the case of female choosiness alone (figure S7; the case of male choosiness alone is not so important because female choosiness is far more abundant biologically and the model implicitly assumes the existence of female choosiness). This implies that the inclusion of male choosiness may destabilize reproductive isolation, which is very counterintuitive. This result is also biologically important because whether male choosiness is possible may depend on the mating system (behavior) of the taxa, so that the relationship between mating system (behavior) and the frequency of speciation can be discussed. I recommend revising the paper emphasizing strongly this important result, which makes the paper more attractive and acceptable to Nature Communications.

Reviewer #3:

Remarks to the Author:

I would first like to apologize if the authors found my language in the earlier review to be harsh. That said, the main problem – that the equations are not consistent with the assumptions – is still there in the revision. Contrary to the authors' response, none of what I said depends on whether or not the loci are physically linked. The point is about what is sometimes called "gametic phase linkage disequilibrium": alleles at different loci are nonrandomly associated in haploid gametes. Sexual selection generates this kind of LD whether or not the loci are physically linked (see Lande 1981, Kirkpatrick 1982, and many later papers). This type of LD is critical to the dynamics in models of sexual selection, for example it is key to Fisher's famous "runaway process". My understanding is that the equations in this ms implicitly assume that there is no gametic phase linkage disequilibrium.

To see the problem from another angle, consider again the example of two loci, A and B. The modeling approach used in the ms keeps track of double heterozygotes (AaBb) with a single frequency. But in fact we need to keep track of two frequencies, those of AB/ab individuals and of Ab/aB individuals. In this notation, st/qv means that alleles st were inherited from one parent and alleles qv were inherited from the other parent. (Again, nothing here depends on physical linkage.) I believe that the equations analyzed in this ms implicitly assume that AB/ab individuals and Ab/aB individuals are equally common. But under the assumptions of this model, those two genotypes will not in fact be equally frequent. Sexual selection generates nonrandom associations between alleles in the gametes that then go on to be the alleles in the offspring that were inherited from each parent. That is true even when the loci are physically unlinked. As the result, selection acting on one locus will cause the other locus to evolve by indirect selection. The equations in this manuscript account for some indirect selection, but because they implicitly ignore gametic phase linkage disequilibrium I believe they give the wrong dynamics.

Response to Referee #1 (Dr. Michael Kopp)

General response:

Again, we thank Dr. Michael Kopp for his very thorough review of our manuscript. In the revised version of the manuscript we aimed at addressing his concerns. We here respond to specific comments that have been raised.

Point-by-point responses to specific comments:

Comment 1: line 72: one-allele mechanisms is unclear here, and might best be simply not mentioned (what is a one-allele mechanisms is the evolution of increased choosiness [in either sex] in both AA and aa subpopulations)

Response 1: Following this suggestion, we do not mention that we model a one-allele mechanism, indeed this helps to avoid any confusion.

Comment 2: line 87 and elsewhere (e.g. 309): Strictly speaking, polygyny does not mean (nor necessarily imply) that all females have equal mating success, but only that one male can fertilize several females.

Response 2: Good point. We now mention that females have equal mating success in “many” polygynous mating systems (e.g., page 4, line 89), and we avoid referring to “a polygyny scenario” when we mention the assumption of equal female mating success.

Comment 3: line 100: Is homotypic preference the same as phenotype matching? If yes, it might be good not to mix these two terms; the first parenthetical in this line might also read “magic trait”

Response 3: We now avoid mixing the terms homotypic preferences and phenotype matching (that refer to the same mating rule). We also refer now to the ecological locus as a so-called ‘magic trait’ (page 5, line 103).

Comment 4: line 120: Without negative frequency-dependence, swamping will occur even under random mating [this sentence and the next one might also simply be deleted]

Response 4: We removed the sentence stating that positive density-dependence causes gene swamping. Instead, we now explain that implementing $s' > 0$ generates negative frequency-dependence, which inhibits gene swamping (page 5, lines 122-124).

Comment 5: line 154: This is true only for weak costs of choosiness!

Response 5: This is now stated in the main text (page 7, lines 156-158).

Comment 6: line 168, one percent of choosy individuals are introduced: does this also hold for the stochastic model, or do you wait for mutations in this case?

Response 6: We initialize simulations with one percent of choosy individuals in all cases, including stochastic simulations. This is to keep the results comparable. Stochastic simulations with no initial introduction of choosy genotypes yield results very comparable to the ones shown, but we believe that presenting the 1% case is better for the sake of consistency.

Comment 7: line 169: I assume also LD between F and M loci, and Hardy-Weinberg at both?

Response 7: Yes, we make this now clear (page 7, lines 172-174).

Comment 8: line 171: is μ the mutation probability per allele or per (diploid) locus? (a common source of confusion!)

Response 8: This is a mutation probability per diploid locus; we have rephrased this sentence so that there is no chance of confusion (page 7, lines 175-176).

Comment 9: line 172, "by averaging over these runs": and over time within runs?

Response 9: Yes, all statistics are calculated over time within runs. This is now made clear in the revised version (page 7, line 178).

Comment 10: lines 174-177: I don't quite understand this argument (see also Fig. S3). In a system with two alleles, heterozygotes always have 50% of heterozygous offspring, no matter what. If they have less mating success, shouldn't they still produce less (homozygous or heterozygous) offspring? Note that ref. [22] (Pennings et al. 2008) uses essentially the same model of female choice, and in that paper, sexual selection favoring heterozygotes can prevent the evolution of choosiness (and hence what you call ecological divergence; indeed, the situation you describe seems to be equivalent to the R/C regime in [22]). I wonder if what happens in your model is the following: You introduce a non-negligible proportion (1%) of choosy females; while almost 50% of the population are still heterozygous at the ecological locus, the choosiness allele rapidly develops strong LD with homozygous genotypes at that locus; in consequence most of the `_choosy_` females actually prefer homozygous males, and this creates sexual selection in favor of homozygotes, and hence does not impede the spread of the choosiness allele.

Response 10: We agree with Dr. Kopp's argument. In our model, sexual selection caused by choosiness cannot develop a *negative* linkage disequilibrium between the choosiness and the ecological loci (e.g., FF and MM associated primarily with Aa). As a consequence of the resulting linkage disequilibrium, sexual selection caused by choosiness ultimately favours homozygotes.

We tried to argue that this effect relies on the genetic basis of the ecological trait that we model. Because heterozygotes Aa always have 50% of heterozygous offspring, sexual

selection cannot develop a negative linkage disequilibrium. This would not occur if mating between individuals with an intermediate ecological trait would result on offspring with an intermediate ecological trait (e.g., if the ecological trait is modelled as a quantitative trait; like in Pennings et al. 2008, thereby leading to the “R/C regime”). In the revised version of the manuscript, we now use Dr. Kopp’s argument (page 7, lines 179-184). Note that this effect does not depend on the frequency of choosy individuals (e.g., when Aa-FF have a high mating success, this never associates with a strong association between genotypes Aa and FF; Fig. S1 and S2).

Comment 11: line 228: Maybe: "In the simplest case, indirect sexual selection on female choosiness may be seen as reflection of direct sexual selection on male choosiness, which is transmitted via LD between the M and F loci. However, this is not necessarily so, as evidenced by parameter regions of female sex. - that overlap with male sex. + in Fig. 2d-f. Indeed, net indirect selection results from LD between all three loci. In particular, choosy males that also carry alleles for female choosiness (genotypes FF-MM) are particularly likely to be homozygous at the ecological locus, whereas choosy males not carrying such alleles (genotypes ff-MM) are somewhat more likely to be heterozygotes Aa [Is this true? I find it hard to judge from Figure S2; also, this Figure does not show results for heterozygous choosiness genotypes Ff or Mm]. While choosy heterozygous males benefit from reduced male-male competition (contributing to sex. + on male choosiness), choosy homozygous males (which are strongly linked to female choosiness) face intensified male-male competition (strongly contributing to sex. - on female choosiness). As a whole, sexual can therefore favor male choosiness and inhibit female choosiness.

Response 11: We rephrased accordingly (page 9, lines 236-245). The LD between genotypes Aa and ff-MM (or FF-mm) is positive when the frequency of choosy females is high. In Fig. S2, we now represent subfigures with a zoomed y-axis to make it clear that these LD are positive. The LD involving heterozygous choosiness genotypes Ff or Mm reflects the one involving homozygous choosiness genotypes ff and mm (e.g., LD of Aa-ff-MM and Aa-Ff-MM is > 0 when there is a high frequency of choosy females, Fig. R1). Therefore, we prefer not to present results on all three-locus linkage disequilibria (this would lead to an unwieldy figure with $3^3=27$ subfigures or additional supplementary figures providing little additional insight). Instead, we now added a brief mention of the relevant results in the caption of Fig. S2.

Figure R1: Genotypic linkage disequilibrium involving genotypes ff-MM and Ff-MM (see caption of Fig. S2 for details). There is a positive genotypic linkage disequilibrium if there is a high frequency of choosy females.

Comment 12: line 348: What exactly is the probability to reach regime P? Do you start repeated simulations in regimes M and FM, and count how many times the simulation enters P? But won't it always reach this regime eventually, if you wait long enough?

Response 12: This corresponds to the mean probability of reaching regime P *at each time step*. In each run, we record how many times the simulation enters regime P given that the system is in regime M or FM. We rephrased this sentence to avoid any confusion (page 16, lines 368).

Comment 13: Fig. S7+8: point out somewhere in the main text that male choosiness alone can evolve only if alpha is very high (about 0.9)

Response 13: This is a good point that we forgot to mention in the main text. Without coevolution of female and female choosiness, the conditions under which male choosiness can evolve are more constrained (page 11, lines 274-277).

Comment 14: line 336 and Fig. S11: I don't get the argument for why dominant choosiness alleles reduce the likelihood of preference cycling. You say that fixation of the dominant allele is difficult, because the recessive allele, once it is rare, is no longer exposed to selection. But at the phenotypic level, this means that the proportion of choosy males becomes very high (my guess is as high as in the recessive model), so selection on female choosiness should vanish, drift should take over etc.

Response 14: Preference cycling is triggered as soon as the fraction of choosy males in the system is very low.

If the allele M coding for male choosiness is dominant, the recessive allele m , once it is rare, is no longer exposed to selection, and remains in the system. As a result, there is a small fraction of nonchoosy males that remain in the system. At the phenotypic level, the proportion of choosy males is high, but not as high as with a recessive M allele.

To illustrate this phenomenon, let us consider the invasion of an advantageous allele A in a standard single-locus diploid model (Figure R2). The initial increase in frequency of a dominant allele A is strong, but the final approach to fixation is more rapid for a recessive allele a . Therefore, after a transition phase (here, 300 time steps) there are more individual expressing the phenotype a if the allele A is dominant than if it is recessive.

We reformulated this sentence to make this argument clearer (page 15, lines 355-357; caption of Fig. S12).

Figure R2: Fixation of an advantageous allele *A* in a simple one-locus diploid model. Allele *A* is either dominant or recessive, and codes for a phenotype **A**, which associates with a selective advantage. If allele *A* is dominant, the fitness coefficients are $W_{AA} = W_{Aa} = \mathbf{W}_A = 1$ and $W_{aa} = \mathbf{W}_a = 1-s$. If allele *A* is recessive, the fitness coefficients are $W_{AA} = \mathbf{W}_A = 1$ and $W_{aa} = W_{Aa} = \mathbf{W}_a = 1-s$. Here, the selection coefficient is $s = 0.1$. The

recursion equation is:
$$p_A(t+1) = \frac{p_A(t)^2 W_{AA} + p_A(t)(1-p_A(t)) W_{Aa}}{p_A(t)^2 W_{AA} + 2p_A(t)(1-p_A(t)) W_{Aa} + (1-p_A(t))^2 W_{aa}}$$

Comment 15: line 341 and Fig. S14: I think I get your argument here: Even if choosiness is not perfect, strong choosiness in combination with other barriers might make reproductive isolation perfect enough to trigger preference cycling. Two comments though: First, I don't understand at all what $\min(P_{\{m,f\}^{\text{male}}}) = \min(P_{\{f,m\}^{\text{female}}})$ means in Fig. S14. Logically, the presence of other barriers should mean that reproductive isolation can never drop below a certain value, even if choosiness evolves to zero, but I don't see how this idea is conveyed by the above condition. Second, the fact that RI cannot drop below a certain value means that the "amplitude" of preference cycles is limited, and hybridisation should stay lower. This effect should be briefly discussed.

Response 15: After thinking about it, we agree that our notation was very confusing. The notation we chose in the previous version of the manuscript could be read as meaning that all $P_{\{m,f\}^{\text{male}}}$ and $P_{\{f,m\}^{\text{female}}}$ reach this minimum value, but that is not the case in those simulations.

If the focal individual is nonchoosy *and* if the genotypes *m* and *f* correspond to a mismatch at the ecological locus, *then* $P_{\{m,f\}^{\text{male}}}$ and $P_{\{f,m\}^{\text{female}}}$ are equal and > 0 . In that case, reproductive isolation can never drop below a certain value. We changed the notation and rephrased the **caption of Fig. S15**.

Additionally, we now briefly discuss the implication of other isolating barriers for the level of reproductive isolation reached during preference cycles as suggested (**page 15, line 363**).

Comment 16: line 391: Incompatibilities can arise in the presence of gene flow if the alleles fixing in the two populations are favored by selection (see Bank, Bürger, Hermisson 2012, Genetics)

Response 16: We did not mean to imply that a perfect premating isolation is required for incompatibilities to be established. Bank et al. (2012) assess the maximum rate of gene flow that allows for the evolutionary origin and the maintenance of incompatibility in parapatry. Therefore, incompatibilities can arise if premating isolation maintains gene flow long enough below this threshold value (i.e., premating is stable). We now refer to “stable premating isolation reducing gene flow” and we cite this important reference (page 17, lines 410-412).

Comment 17: Appendix A: I don't quite understand the second-to-last paragraph, in particular the part about cheating and positive frequency-dependence (see also my editorial comments below).

Response 17: We follow the suggestion here. Describing coevolutionary dynamics that we were expecting but that did not occur is probably not particularly useful, and removing it helps to shorten the text.

Comment 18: Fig. S7 shows that male choosiness alone can evolve only for large values of alpha. This should be pointed out in the main text (when describing Fig. 1).

Response 18: See Response 13.

Editorial comments:

We rephrased many sentences as suggested. Here we provide additional information concerning specific editorial comments.

line 213: Should the dashed arrows in Fig. 1c point in both directions (because of the vice versa)?

=> Indirect selection acts in both directions (that's why we wrote “and vice-versa”), but we only represent the indirect selective forces that act on female choosiness through their effect on male choosiness (pink, red and blue arrows). That's why the dashed arrows do not point in both directions. We removed “and vice-versa” in the main text (page 8, line 220) when we describe the figure

line 219: I don't see any arrows with low opacity in fig. 1c. Is this sentence still relevant?

=> We increased transparency of those arrows to make it clearer in Fig. 1C

- Fig. S6: We compute the F_{st} statistic as a measure of neutral genetic differentiation between ecotypes ... male and female choosiness favour ... genetic differentiation is stable [I suppose heterozygotes are ignored for this calculation?]

=> Indeed, heterozygotes are ignored for this calculation.

- 223: the notation $\text{sex. } +$ and $\text{sex. } -$ looks really weird in the main text; can you change it to something like $S+$ and $S-$ or $\text{sex}+$ and $\text{sex}-$ and maybe italicize it or put it in quotes?

=> We take the point; we removed these notations both in **Fig. 2** and in the main text. Indeed this helps to avoid any confusion.

Response to Referee #2

General response:

We thank Referee #2 for reviewing our manuscript. We are happy that s/he found our argument clearer and recommended our paper for publication after revision. We hope that with the following amendments we have addressed his/her concerns.

Point-by-point responses to specific comments:

Comment 1:

I previously commented that the argument did not correspond to the analysis. In the revised manuscript, the argument becomes clearer and I realized that I somewhat misunderstood the argument. But therefore, I also realized that I somewhat overestimated the value of the paper, because I previously thought that the paper showed that mutual mate choice is less effective for reproductive isolation than mate choice by single sex. The authors merely argue that reproductive isolation is sometimes destabilized by mutual mate choice. Since reproductive isolation is easily destabilized when preference loci disappear, which may occur by genetic drift, the result is not impressive enough. The authors also emphasize that preference cycling occurs when considering mutual mate choice, but this result is theoretically not surprising because it occurs in simpler sexual selection models, and biologically not significant enough because no empirical data show this phenomenon (if existed, please raise a reference).

Nevertheless, I recommend this paper for publication after a revision, because this paper seems to show that mutual mate choice is less effective for reproductive isolation than the case of female choosiness alone (figure S7; the case of male choosiness alone is not so important because female choosiness is far more abundant biologically and the model implicitly assumes the existence of female choosiness). This implies that the inclusion of male choosiness may destabilize reproductive isolation, which is very counterintuitive.

Response 1:

Thank you for this new evaluation. We agree with its gist, but would also like to point out that our results do not only demonstrate the fact that “reproductive isolation is easily destabilized when preference loci disappear, which may occur by genetic drift”. More importantly, we describe a variety of coevolutionary phenomena that do not occur in simpler sexual selection models (that involve single-sex choice). Indeed, in our simulations, female mate choice is not easily destabilized if it evolves alone (**Fig. S8 and S9**, and as raised by Referee #2).

We show that (1) the evolution of male choosiness makes female choosiness less robust to genetic drift, and (2) the loss of female choosiness (by drift) triggers the loss of male choosiness by selection. When we account for preference coevolution, the interplay between the selection pressures acting on female and male choosiness is rather complex, and the coevolutionary dynamics presented in our paper is triggered by drift (the loss of female choosiness), and completed by selection (the loss of male choosiness) (**page 3, lines 66-68; page 13, line 323-324; page 17, line 406-408**). We now emphasize this point in the abstract (**page 1, lines 9-10**).

This coevolutionary dynamics explains why “mutual mate choice is less effective for reproductive isolation than the case of female choosiness alone”, which is the very

counterintuitive result of our paper (as emphasized now in the abstract, page 1, lines 10-11, and in the Discussion, page 17, line 400).

Unfortunately, in our knowledge, no empirical data show this phenomenon. The problem is that no study has characterized the strength of isolating barriers at a sufficient temporal scale. Our study therefore belongs to the category of theory that hopefully inspires new ways to look at empirical patterns, rather than being able to dig into existing datasets to explain them (page 18, lines 419-427). We hope that our study will stimulate such empirical research.

Comment 2:

This result is also biologically important because whether male choosiness is possible may depend on the mating system (behavior) of the taxa, so that the relationship between mating system (behavior) and the frequency of speciation can be discussed. I recommend revising the paper emphasizing strongly this important result, which makes the paper more attractive and acceptable to Nature Communications.

Response 2:

We agree wholeheartedly with Referee #2. Our model suggests that the link between mating systems and speciation may not be as straightforward as previously thought. Species with a mating system that is prone to the evolution of male mate choice (in addition to female mate choice) may not necessarily associate with a high speciation rate, because of the 'preference cycling' dynamics we highlight in our manuscript.

In a new paragraph of the Discussion, we now emphasize that our predictions suggest that the mating system of the taxa may associate with the frequency of speciation in quite counterintuitive ways (page 18, lines 438-444). We also emphasize this point in our Conclusion (page 19, line 474).

Response to Referee #3

General response:

Again, we are very thankful to Referee #3 for reviewing our manuscript. By representing the gamete phase disequilibrium in a new supplementary figure, we hope that we addressed the Referee #3's main technical concern.

Comment:

I would first like to apologize if the authors found my language in the earlier review to be harsh. That said, the main problem – that the equations are not consistent with the assumptions – is still there in the revision. Contrary to the authors' response, none of what I said depends on whether or not the loci are physically linked. The point is about what is sometimes called "gametic phase linkage disequilibrium": alleles at different loci are nonrandomly associated in haploid gametes. Sexual selection generates this kind of LD whether or not the loci are physically linked (see Lande 1981, Kirkpatrick 1982, and many later papers). This type of LD is critical to the dynamics in models of sexual selection, for example it is key to Fisher's famous "runaway process". My understanding is that the equations in this ms implicitly assume that there is no gametic phase linkage disequilibrium.

To see the problem from another angle, consider again the example of two loci, A and B. The modeling approach used in the ms keeps track of double heterozygotes (AaBb) with a single frequency. But in fact we need to keep track of two frequencies, those of AB/ab individuals and of Ab/aB individuals. In this notation, st/qv means that alleles st were inherited from one parent and alleles qv were inherited from the other parent. (Again, nothing here depends on physical linkage.) I believe that the equations analyzed in this ms implicitly assume that AB/ab individuals and Ab/aB individuals are equally common. But under the assumptions of this model, those two genotypes will not in fact be equally frequent. Sexual selection generates nonrandom associations between alleles in the gametes that then go on to be the alleles in the offspring that were inherited from each parent. That is true even when the loci are physically unlinked. As the result, selection acting on one locus will cause the other locus to evolve by indirect selection. The equations in this manuscript account for some indirect selection, but because they implicitly ignore gametic phase linkage disequilibrium I believe they give the wrong dynamics.

Response:

We welcome the new tone of this conversation. Unfortunately, we are still in disagreement with the argument raised by Referee #3. As argued in our response following the first round of review, there is no need to keep track of the genotypes of haploid gametes because there is free recombination in our model (i.e., no physical linkage between loci).

One point raised by Referee #3 may explain where our disagreement comes from. Referee #3 suspects that our equations "implicitly assume AB/ab individuals and Ab/aB individuals are equally common" in the case of double heterozygotes. This is not the case. Because those individuals produce exactly the same gametes (because of free recombination), we do not keep track of their relative frequencies ($\text{freq}(\text{AB}/\text{ab}) / (\text{freq}(\text{AB}/\text{ab}) + \text{freq}(\text{Ab}/\text{aB}))$ and $\text{freq}(\text{Ab}/\text{aB}) / (\text{freq}(\text{AB}/\text{ab}) + \text{freq}(\text{Ab}/\text{aB}))$). In other words,

our recursion equations simulate the change in frequency of $\text{freq}(AaBb) = \text{freq}(AB/ab) + \text{freq}(Ab/aB)$, but do not imply that $\text{freq}(AB/ab) = \text{freq}(Ab/aB)$.

Referee #3 states that our model “implicitly assume that there is no gametic phase disequilibrium”. In reality, our recursion equations generate such gametic phase disequilibrium where alleles at different loci are nonrandomly associated in haploid gametes. Following the editor’s suggestion, we now represent the gametic phase disequilibrium that emerges in our simulations (**Fig. S3**) and we mention it in the main text (**page 8, lines 189-190**). We hope that it will avoid readers to misunderstand our modelling approach. The code we used is now made available for peer-review and for any subsequent reader wishing to check what is happening to the relevant frequencies.

Reviewers' Comments:

Reviewer #1:

Remarks to the Author:

I am very happy with this second revision, and I only have one more minor comment (plus some editorial suggestions). I recommend the paper for publication after accounting for the points below. I do not need to see it again. I add that I completely agree with the authors' response to reviewer three.

Minor comments:

I think there is still a slight confusion as to why, in the present model, female choosiness does not impede ecological divergence (line 179ff, see also response 10 to my comment in the rebuttal letter, and Fig. S4). The authors argue that this is because heterozygotes Aa do not breed true and always produce 50% of non-heterozygote offspring; in addition, they adopted my argument that the MM genotype develops linkage disequilibrium with homozygous ecological genotypes AA and aa . This is all true, but I think the authors misunderstood what happens in the Pennings et al. 2008 model (and others, such as Matessi et al. 2001), where sexual selection due to female choosiness can impede initial divergence or cause the evolution of choosiness to stop at an intermediate level. Like the present model, the Pennings et al. model assumes that the ecological trait is governed by a single diallelic locus (so it is not a quantitative trait, either). It also is not ecological divergence per se that is hampered by sexual selection (as would be the case if heterozygotes could breed true), but the evolution of choosiness in small steps. That is, positive frequency-dependent sexual selection increases the mating success of heterozygotes (regardless of offspring genotype), and this counterbalances the negative frequency-dependent ecological selection that causes heterozygotes to have reduced viability. As a consequence, the invasion fitness gradient for choosiness may become negative, that is, increased choosiness is selected against. In the R/C and P/C regimes, this creates an unstable intermediate equilibrium for choosiness, which cannot be surpassed as long as choosiness evolves in small steps (as is the typical adaptive-dynamics assumption). However, I think the key difference between this situation and the present model is that, in the present model, choosiness evolves in a single, very large step, which allows the population to "jump" over the unstable equilibrium (thanks to the above mentioned LD). This "jumping" mechanism has also been shown in simulations by Pennings et al., Fig. 6, and Rettelbach et al. 2011. (Another mechanism shown by Pennings et al., Kirkpatrick and Nuismer, and Bürger and Schneider is that positive frequency-dependent sexual selection causes collapse of the ecological polymorphism. This is precluded

by the factor s' in the present model.) Since this argument is rather subtle, I think that the paragraph line 179ff should probably be moved to either the Results (e.g., after l. 201) or Discussion and modified accordingly. E.g. along the lines "Note that in previous models [citations], sexual selection created by female choosiness has been shown to impede speciation if the initial population was close to random mating (such that intermediate phenotypes had the highest mating success) and choosiness evolves in small steps. This is not the case in our model, where choosiness jumps from zero to almost one. In this, a strong linkage disequilibrium develops between MM genotypes and AA or aa genotypes, such that choosy females are mostly ecological homozygotes, increasing the mating success of homozygous males and indirectly favoring an increase in the frequency of the MM genotype." The caption of Fig. S4 might also need to be adjusted.

Thanks for Figure R2, I was not at all aware of this phenomenon.

Editorial comments:

- 5: or "coevolve under a phenotype matching rule"
- 11: both male and female mate choice
- 23: When these traits are simultaneously are under divergent or disruptive ecological selection (so-called magic traits [Gavrilets 2004, Servedio et al. 2011])...
- 25: [Not sure I would start a new paragraph here.]
Theoretically, this effect creates indirect selection for a further increase in choosiness (...), establishing or strengthening premating isolation ...
- 26: You might try to merge these two sentences, e.g., "... homotypic preferences often induce positive frequency-dependent sexual selection, since as long as assortment is not perfect, individuals having the most common phenotype enjoy the highest mating success."
- 28: This is only true for a magic trait. Maybe something along these lines: Once diverging populations are sufficiently differentiated with respect to a magic trait, disruptive ecological selection on that trait is therefore complemented by disruptive sexual selection due to choosiness, which in turn may drive the evolution of even stronger choosiness. (But (but I am not quite happy with this version, either.)
- 32: are, indeed, often associated
- 35: Previous theoretical developments
- 36: In principle, however, the indirect selection effect explained above (i.e., reduced production of unfit intermediate offspring) should favor increased choosiness in both males and females, even though evolutionary pressures acting on the two sexes are profoundly different.
- 71: but without mate choice, ecological divergence is hampered

by random mating that brings ...

- 74: or "the trait determined by locus A"
- 89: as is the case in many polygynous mating systems [citations would be nice here]
- 118: delete "ensuring the maintenance of polymorphism"
- 141: comma before and
- 158: whereas female choosiness does not evolve...
- 176: thanks for clarifying that μ is mutation rate per diploid locus; however, this somewhat contradicts the formulation in line 168, where each allele is said to mutate with probability μ
- 228: from the combined viability and sexual selection
- 229: of sexual selection alone
- 240: The reason is that net indirect selection ...
- 245: and simultaneously inhibit
- 267: I would move ($s < 0.05$) to after "When viability selection is weak"); same in . 273
- 267: do not reallocate courtship effort at all
- 280: Finally, changing the dominance hierarchy (i.e., making alleles F or M dominant)...
- 325: do not all have
- 356: is less rapid (results not shown);
- 389: or maybe: when they are mostly being courted by males of their own ecotype
- 393: females
- 411: I would omit "reducing gene flow" here
- 419: remove "so-called" here, since speciation continuum has already been mentioned in line 414)
- 423: maybe "whether gene flow will be reduced"
- 428: mate preferences
- 432: use single quotes for consistency
- Some suggestions for the paragraph starting on line 428: ... By treating the reallocation of courtship effort as a parameter(α), our model covers a wide variety of courtship and mating systems in animals. In particular, reallocation seems plausible if 'courtship effort' refers to time (e.g., for mate searching or performing complex displays 37) or energy (e.g., for resources-demanding spermatophores 35 or nuptial gifts 36). In contrast, reallocation will be effectively reduced if searching for more preferred mates entails mortality costs. Our results suggest that obtaining estimates of courtship reallocation in nature will be an important ingredient for increasing our understanding of divergence in the presence of gene flow. In particular, we predict that mutual mate choice should be favored whenever choosy males can reallocate even a small proportion of courtship effort towards more preferred females, but that it is precisely such partial reallocation that can induce preference cycling, possibly limiting divergence.
- 439: add "(in agreement with our predictions)" to the end of this sentence

- 441: in determining speciation rate (or: in determining the likelihood of speciation)
- 442: between mating system and speciation
- 444: with high speciation rates
- 447: choosy individuals
- 442: and preventing
- 457: I would not put "favour" in quotes here
- 467: effort
- 470: multiple potential barriers to gene flow
- Appendix B, first line: comma before "and choosiness"
- Figure S1, end of blue part: maybe "and are not shown either"
- Fig. S2, line 6: linkage disequilibrium; somewhere in the middle: under these conditions; last sentence: This is why
- Fig. S3, line 4: linkage disequilibrium; later: associate with both homozygous ecotypes
- Fig. S7: In offspring, each neutral allele can mutate ... to any other allele.

Signed,

Michael Kopp.

Response to Referee #1 (Dr. Michael Kopp)

General response:

We thank Dr. Michael Kopp for another very thorough review of our manuscript. Since the first submission, the comments and editorial suggestions of Dr. Michael Kopp, among others, really helped us to produce a much better paper. In this revised version of the manuscript we aimed at addressing his last minor comment. We also followed his more minor editorial suggestions.

Response to specific comment:

Comment: I think there is still a slight confusion as to why, in the present model, female choosiness does not impede ecological divergence (line 179ff, see also response 10 to my comment in the rebuttal letter, and Fig. S4). The authors argue that this is because heterozygotes Aa do not breed true and always produce 50% of non-heterozygote offspring; in addition, they adopted my argument that the MM genotype develops linkage disequilibrium with homozygous ecological genotypes AA and aa. This is all true, but I think the authors misunderstood what happens in the Pennings et al. 2008 model (and others, such as Matessi et al. 2001), where sexual selection due to female choosiness can impede initial divergence or cause the evolution of choosiness to stop at an intermediate level.

Like the present model, the Pennings et al. model assumes that the ecological trait is governed by a single diallelic locus (so it is not a quantitative trait, either). It also is not ecological divergence per se that is hampered by sexual selection (as would be the case if heterozygotes could breed true), but the evolution of choosiness in small steps. That is, positive frequency-dependent sexual selection increases the mating success of heterozygotes (regardless of offspring genotype), and this counterbalances the negative frequency-dependent ecological selection that causes heterozygotes to have reduced viability. As a consequence, the invasion fitness gradient for choosiness may become negative, that is, increased choosiness is selected against. In the R/C and P/C regimes, this creates an unstable intermediate equilibrium for choosiness, which cannot be surpassed as long as choosiness evolves in small steps (as is the typical adaptive-dynamics assumption).

However, I think the key difference between this situation and the present model is that, in the present model, choosiness evolves in a single, very large step, which allows the population to "jump" over the unstable equilibrium (thanks to the above mentioned LD). This "jumping" mechanism has also been shown in simulations by Pennings et al., Fig. 6, and Rettelbach et al. 2011. (Another mechanism shown by Pennings et al., Kirkpatrick and Nuismer, and Bürger and Schneider is that positive frequency-dependent sexual selection causes collapse of the ecological polymorphism. This is precluded by the factor s' in the present model.)

Since this argument is rather subtle, I think that the paragraph line 179ff should probably be moved to either the Results (e.g., after l. 201) or Discussion and modified accordingly. E.g. along the lines "Note that in previous models [citations], sexual selection created by female choosiness has been shown to impede speciation if the initial population was close to random mating (such that intermediate phenotypes had the highest mating success) and choosiness evolves in small steps. This is not the case in our model, where choosiness jumps from zero to almost one. In this, a strong linkage disequilibrium develops between MM genotypes and AA or aa genotypes, such that choosy females are mostly ecological homozygotes, increasing the mating success of homozygous males and indirectly favoring an increase in the frequency of the MM genotype." The caption of Fig. S4 might also need to be adjusted.

Response: We indeed had misunderstood the process underlying the effect of sexual selection on the evolution of choosiness and on ecological divergence in those papers. We

thank Dr. Michael Kopp for explaining in great detail why the effect of sexual selection on ecological divergence differs between papers. In the revised version of the manuscript, we briefly explain why the evolution of choosiness does not impede ecological divergence in our model by following the suggested paragraph of Dr. Michael Kopp.

Given that this phenomenon is somehow unrelated to the process of preference cycling that we describe in our manuscript, we fear that moving this part in the Results/Discussion would confuse the reader. Unless our Editor insists, we would prefer to keep this statement at the end of the Methods to preserve the flow of the main text.